# Somatic and vicarious pain are represented by dissociable multivariate brain patterns

**Anjali Krishnan[1,2,3]\*, Choong-Wan Woo[1,2†], Luke J Chang[1,2,4†], Luka Ruzic[1,2,5], Xiaosi Gu[6,7], Marina López-Solà[1,2], Philip L Jackson[8], Jesús Pujol[9], Jin Fan[10,11], Tor D Wager[1,2]\***

[1]Institute of Cognitive Science, University of Colorado Boulder, Boulder, United States; [2]Department of Psychology and Neuroscience, University of Colorado Boulder, Boulder, United States; [3]Department of Psychology, Brooklyn College of the City University of New York, Brooklyn, United States; [4]Department of Psychological and Brain Sciences, Dartmouth College, Hanover, United States; [5]Center for Cognitive Neuroscience, Duke University, Durham, United States; [6]Wellcome Trust Centre for Neuroimaging, University College London, London, United Kingdom; [7]School of Behavioral and Brain Sciences, The University of Texas at Dallas, Richardson, United States; [8]École de Psychologie, Université Laval, Quebec City, Canada; [9]MRI Research Unit, Radiology Department, Hospital del Mar, CIBERSAM G21, Barcelona, Spain; [10]Department of Psychology, Queens College of the City University of New York, New York City, United States; [11]Department of Psychiatry and Neuroscience, Icahn School of Medicine at Mount Sinai, New York City, United States

**\*For correspondence:** anjali. krishnan05@brooklyn.cuny.edu (AK); tor.wager@colorado.edu (TDW)

†These authors contributed equally to this work

**Abstract** Understanding how humans represent others' pain is critical for understanding pro-social behavior. 'Shared experience' theories propose common brain representations for somatic and vicarious pain, but other evidence suggests that specialized circuits are required to experience others' suffering. Combining functional neuroimaging with multivariate pattern analyses, we identified dissociable patterns that predicted somatic (high versus low: 100%) and vicarious (high versus low: 100%) pain intensity in out-of-sample individuals. Critically, each pattern was at chance in predicting the other experience, demonstrating separate modifiability of both patterns. Somatotopy (upper versus lower limb: 93% accuracy for both conditions) was also distinct, located in somatosensory versus mentalizing-related circuits for somatic and vicarious pain, respectively. Two additional studies demonstrated the generalizability of the somatic pain pattern (which was originally developed on thermal pain) to mechanical and electrical pain, and also demonstrated the replicability of the somatic/vicarious dissociation. These findings suggest possible mechanisms underlying limitations in feeling others' pain, and present new, more specific, brain targets for studying pain empathy.

**eLife digest** The ability to experience others' pain is a cornerstone of empathy, and binds us together in times of hardship. However, we have not yet fully understood the complex interactions in the brain that make people empathetic to others' suffering. One possibility is that we experience others' pain through the activation of the same brain regions as those that enable us to experience physical pain ourselves.

To test this idea, Krishnan et al. compared patterns of brain activity in human volunteers as they experienced pain (from heat being applied to their forearm or foot) or watched images of others' hands or feet being injured. While watching these images, the volunteers were asked to try to imagine that the injuries were happening to their own bodies.

The patterns of brain activity that arose when the volunteers observed someone else in pain did not overlap with the patterns produced when the volunteers experienced pain themselves. Instead, seeing someone else in pain activated regions involved in taking another person's perspective. This process, which is known as mentalizing, involves thinking about the other person's thoughts, intentions and preferences. Thus within the brain, the experience of observing someone else in pain is distinct from that of experiencing physical pain in oneself.

The results presented by Krishnan et al. raise new questions about how the brain regions involved in empathy help us to relate to other people when they experience different types of pain. Future studies should explore the factors that influence our ability to adopt another's perspective, and whether it might be possible to improve this ability.

# Introduction

"Though our brother is upon the rack, as long as we ourselves are at our ease, our senses will never inform us of what he suffers. . . . it is by the imagination only that we can form any conception of what are his sensations."

Adam Smith (1759)

A fundamental feature of social interactions is our capacity for vicarious experience—the ability to perceive another person's affective state, reference it to ourselves, and generate an emotional response. This ability provides the foundation for empathy and cooperative behavior (*Smith, 1759*) by allowing us to recognize and respond to suffering in others (*Batson et al., 1981*), and learn from their experiences (*Olsson and Phelps, 2007*). Vicarious experiences of others' pain, in particular, aid in representing others' distress with sufficient vividness and importance that we are moved to action.

But how do we represent others' pain, and how vivid and automatic are such representations? Theories differ markedly on this point. One theory suggests that vicarious pain involves *shared experience*, activating neural circuits that represent somatic pain in the perceiver. Such theories are based largely on overlapping activity in the dorsal anterior cingulate cortex (dACC) and anterior insular (aINS) cortices when both experiencing pain and observing pain in others (*Corradi-Dell'Acqua et al., 2011*; *2016*; *Fan et al., 2011*; *Jackson et al., 2005*; *2006a*; *Lamm et al., 2007*; *2011*; *Ogino et al., 2007*; *Singer et al., 2004*). Such overlaps have also been found in primary and secondary somatosensory areas [i.e., SI and SII] (*Jackson et al., 2005*; *Keysers et al., 2004*; *Lamm et al., 2011*; *Singer et al., 2004*). Witnessing others in pain can increase one's own pain (*Langford et al., 2006*; *Loggia et al., 2008*), and placebos given for somatic pain can also reduce vicarious 'pain' (*Rütgen et al., 2015*). Together, these findings have been taken as brain evidence for shared representation of pain. In particular, dACC and aINS activation are thought to reflect a kind of shared experience that has been described in terms of 'neural resonance' (*Zaki et al., 2012*), and as 'automatic' (*Singer et al., 2004*). By this account, empathy is 'built *bottom up* from relatively simple mechanisms of action production and perception' (*Iacoboni, 2009*).

Another theory suggests that vicarious pain may be primarily a reflective, cognitive experience whose experiential qualities are hard to mimic or simulate directly (*Hooker et al., 2008*;

*Loewenstein, 1996*; *Morley and Morley, 1993*; *Zaki, 2014*). Others' pain need not activate somatic pain representations to be aversive (*Chang et al., 2015*), and empathy may work by engaging other emotional systems apart from 'pain in oneself' (*Hooker et al., 2008*). Unlike somatic pain, vicarious pain is all too easy to ignore. For example, many participants will inflict substantial pain on an innocent person when instructed to do so by authority figures (*Cheetham et al., 2009*; *Milgram, 1963*). We also ignore our own past and future pain (*Gilbert et al., 1998*; *Loewenstein, 1996*; *Van Boven and Loewenstein, 2003*), making decisions in line with our goals when pain is distant but often reversing those choices when pain is imminent.

Brain evidence also suggests that vicarious pain may rely on circuits specialized for representing the thoughts and intentions of others (*Cheetham et al., 2009*; *Zaki et al., 2007*; *2009*). A network encompassing the dorso-medial prefrontal cortex [dmPFC], posterior cingulate cortex [PCC], temporal-parietal junction [TPJ], and superior temporal sulcus [STS; (*Frith and Frith, 2006*; *Van Overwalle, 2009*; *Zaki et al., 2012*)] is reliably involved in 'mentalizing'—thinking about others' thoughts (*Saxe and Kanwisher, 2003*), preferences (*Mitchell et al., 2006*), and intentions (*Hampton et al., 2008*)—and in imagining one's own and others' responses to painful stimuli (*Jackson et al., 2006a*; *Lamm et al., 2011*). This 'mentalizing' network is distinct from both pain-related circuitry and circuitry thought to underlie shared affective experiences and actions, including the dACC and aINS (*Frith and Frith, 2006*; *Iacoboni, 2009*; *Shamay-Tsoory et al., 2009*; *Singer et al., 2004*; *Van Overwalle and Baetens, 2009*; *Zaki et al., 2007*; *2012*).

A critical question in assessing shared representation is whether the overlapping brain activity observed really reflects shared *pain-related* processes. Several recent lines of evidence suggest this may not be the case. In a meta-analysis of over 3500 neuroimaging studies available from Neurosynth.org, activation in the aINS and dACC were among the most frequently observed findings across all kinds of tasks (*Yarkoni et al., 2011*), suggesting that activation in these regions frequently has nothing to do with pain. It is possible that activity in portions of the dACC is *preferentially* related to pain on average; for example, *Lieberman and Eisenberger, 2015* used Neurosynth.org to identify a statistical association between studies using the term 'pain' and activation of the dACC. However, this does not mean that dACC activity is sufficient to infer pain, as the dACC responds to a variety of cognitive and emotional events that are not painful (*Wager et al., 2016*). For instance, electrophysiological and optogenetic studies have identified neurons engaged during foraging behavior, attention, emotion, reward expectancy, skeletomotor and visceromotor activity, and other functions (*Davis et al., 2005*; *Picard and Strick, 1996*; *Shidara and Richmond, 2002*). 'Pain-encoding' portions of the dACC can be activated in individuals with a congenital insensitivity to pain (*Salomons et al., 2016*). Only a small minority of dACC neurons are pain-related (*Davis et al., 2005*; *Hutchison et al., 1999*), and the dACC encodes emotional events, including rejection and general negative emotion, in a way that is distinct from how it encodes pain (*Chang et al., 2015*; *Woo et al., 2014*). More fine-grained analyses of population-level representations are required to make inferences about pain based on activity in the dACC and other regions.

In this study, we attempted to identify representations (or markers) for somatic and vicarious pain—both within the dACC and aINS and across the brain—and test their similarity. For a marker (e.g., a multivariate brain pattern) to serve as a representation of pain, it should satisfy three criteria. It should: a) closely track pain experience (be *sensitive* to the presence of pain); b) not respond to experiences that are defined as non-painful (be pain-*specific*); and c) generalize across multiple forms of pain (*Woo and Wager, 2015*). Only to the degree that brain patterns *represent* somatic and vicarious pain does testing their similarity bear on shared representation theories. Previous studies [e.g.,(*Corradi-Dell'Acqua et al., 2011*; *Corradi-Dell'Acqua et al., 2016*)] have compared the similarity of self-pain and other-pain patterns, but these patterns have not shown strong sensitivity to pain experience [criterion (*a*) above]. Indeed, prior work has suggested that sensitivity to pain and emotional experiences requires combining signals across brain regions and networks (*Chang et al., 2015*; *Kassam et al., 2013*; *Kragel and Labar, 2013*; *Nummenmaa et al., 2014*; *Wager et al., 2015*), whereas previous studies of vicarious pain have focused only on local patterns within a single region at a time. In addition, the patterns identified as shared across self- and other-pain have responded to other types of negative emotion (e.g., emotional pictures, disgust, and unfairness), and so do not satisfy criterion (*b*) for pain representations. Finally, no vicarious pain studies have tested generalizability of a specific brain pattern across types of pain [criterion (*c*)].

Here, in Study 1, we used a 3 (stimulation level) × 2 (body site) × 2 (pain modality; i.e., somatic vs. vicarious) experimental design, with pain reports after each trial, to identify patterns that predicted the level of reported somatic and vicarious pain in response to painful heat and observation of pain in others, respectively. Between-participant machine learning analyses were used to identify patterns that test *sensitivity* [i.e., satisfy criterion (*a*) above] for both pain modalities (i.e., somatic vs. vicarious), and to provide an unbiased test of similarity and cross-prediction across somatic and vicarious pain predictive patterns. In addition, we tested somatotopy (upper versus lower limb) within sensory cortices and other systems for both pain modalities, and compared the somatotopic representations. The identification of patterns that can be generalized across participants allowed us to test the specificity [criterion (*b*)], and generalizability [criterion (*c*)] of these patterns in additional studies.

Previous work has identified a pattern that is, thus far, sensitive and specific [criteria (*a*) and (*b*) above] for somatic pain across multiple studies. We used this pattern, called the Neurologic Pain Signature [NPS; (*Wager et al., 2013*)], in our primary analyses, and attempted to identify a parallel pattern for vicarious pain. The NPS has over 90% sensitivity and specificity in predicting somatic pain relative to several other salient states, including non-painful warmth, anticipated pain, pain recall (*Wager et al., 2013*), social rejection (*Woo et al., 2014*), and general negative emotion (*Chang et al., 2015*). In the current paper, Study 2 and Study 3 tested the generalizability of the NPS to mechanical and electrical pain, respectively [addressing criterion (*c*) above]. Study 3 also provided a replication of the sensitivity, specificity, and similarity of somatic and vicarious pain predictive patterns. Our primary goal was to identify—and test the similarity of—whole-brain patterns for somatic and vicarious pain that generalize across individuals and can be tested prospectively in multiple studies. However, we also conducted analyses focused on the dACC and aINS specifically, and on pain-predictive patterns identified for each individual participant. In addition, we conducted 'searchlight' analyses designed to test the maximal similarity of local brain patterns related to somatic and vicarious pain across the brain, and provide inferences on whether distributed, whole-brain patterns are necessary to capture representations of both somatic and vicarious pain.

## Results

### Reported intensity

In Study 1, for the *somatic pain* condition, participants were scanned using functional Magnetic Resonance Imaging (*f*MRI) while they experienced three levels of thermal pain (46, 47 or 48°C; *Figure 1A*) on their left volar forearm (the 'upper limb or UL' site) and left dorsal foot (the 'lower limb or LL' site), and rated the intensity of pain with their right hand following every trial. In the *vicarious pain* condition, participants were scanned while they viewed images depicting injury to others' right hands (the 'upper limb or UL' site) and feet (the 'lower limb or LL' site) and engaged in perspective-taking (*Jackson et al., 2006a*; *2005*)—participants imagined that the injuries were happening to their own bodies—to actively reference the observed pain to their own bodies.

Participants reported increased intensity with increasing levels of somatic pain for both upper limb and lower limb sites with no significant difference between body sites ($t_{UL}(27) = 12.93$ p<0.0001, $t_{LL}(27) = 10.36$, p<0.0001, $t_{UL-LL}(27) = 1.08$, n.s.; *Figure 1B*). Participants also reported increasing intensity with increasing levels of vicarious pain for both upper limb and lower limb sites, with no significant difference between sites ($t_{UL}(27) = 11.99$, p<0.0001, $t_{LL}(27) = 11.37$, p<0.0001, $t_{UL-LL}(27) = 1.49$, n.s.; *Figure 1B*). Intensity ratings were robust and comparable for both somatic and vicarious pain conditions, with no significant difference between mean behavioral ratings for both types of experiences ($t_{Som-Vic}(27) = -0.05$, n.s).

### The NPS is sensitive and specific to somatic pain

To test whether activity in the NPS pattern (*Figure 2A*) tracked both somatic and vicarious pain intensity, we calculated the NPS pattern response [the weighted average activation; (*Wager et al., 2013*)] for each single-participant activation map (regression parameter estimate maps from single-participant general linear models) for each condition (3 stimulation levels × 2 body sites × 2 pain modalities [i.e., somatic vs. vicarious]). This provided a measure of the NPS pattern activation in each

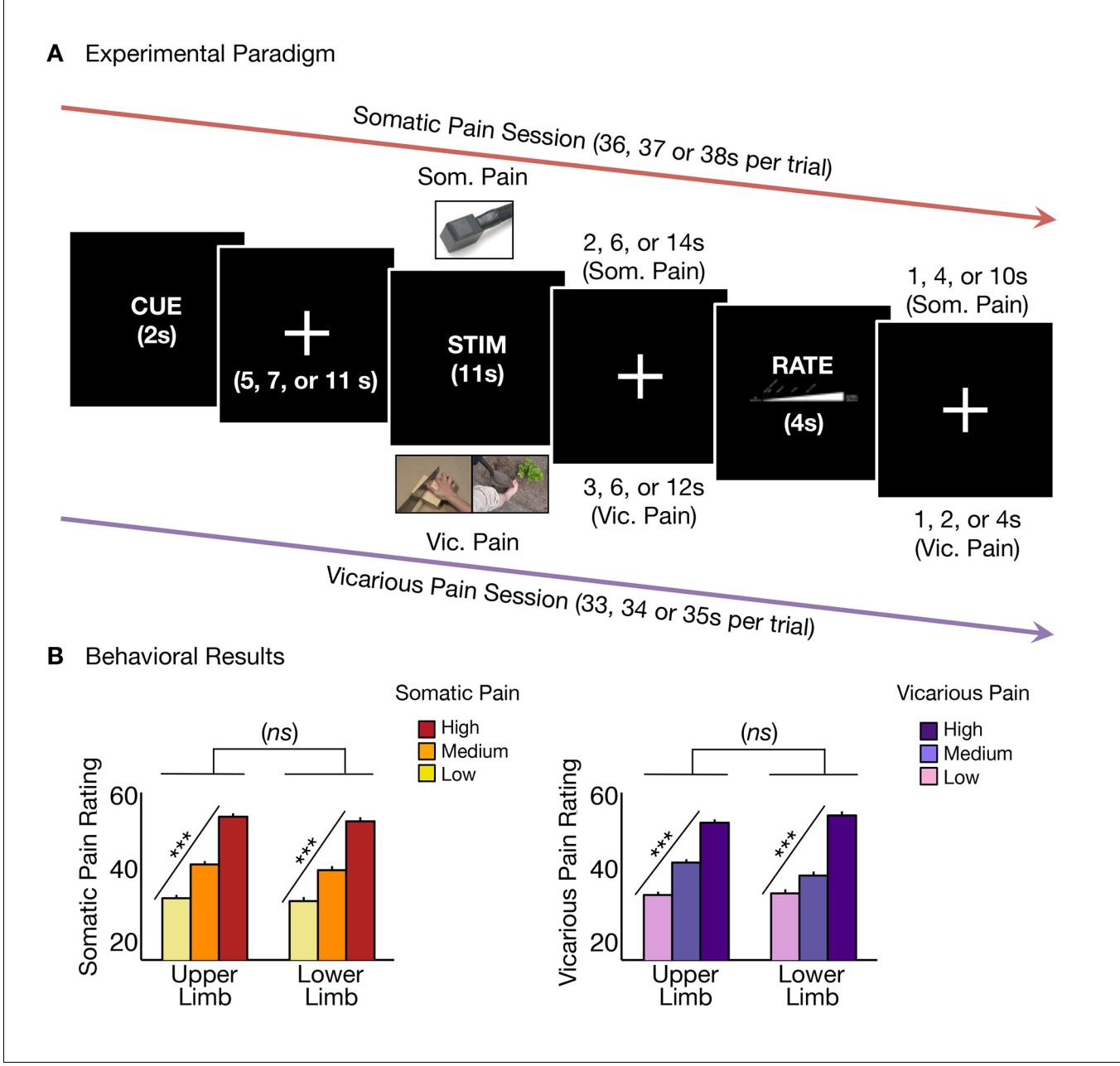

**Figure 1.** Experimental paradigm and behavioral results. (A) Trial timeline for somatic and vicarious pain sessions (differing only by stimulation type and fixation jitter, see 'Materials and methods' for more details); (B) Averaged behavioral ratings for somatic and vicarious pain across levels of stimulation and body site (i.e., upper limb and lower limb; including within-participant standard error of the mean).

condition for each participant, which we analyzed for effects of stimulation level within each body site and for differences across body sites and pain modalities.

The NPS response increased monotonically for each level of somatic pain for both upper limb and lower limb sites ($t_{UL}(27) = 9.08$, $p<0.0001$, accuracy$_{High-Low}= 100\%$, $p<0.0001$; $t_{LL}(27) = 8.88$, $p<0.0001$, accuracy$_{High-Low} = 100\%$, $p<0.0001$; see *Figure 2B and C*. The NPS response was slightly reduced on the lower limb site ($t_{UL-LL}(27) = 2.13$, $p<0.05$; see *Figure 2B and C*), but this difference was not significant after controlling for rated intensity, implying that the NPS response magnitude is consistent with the reported intensity across both sites.

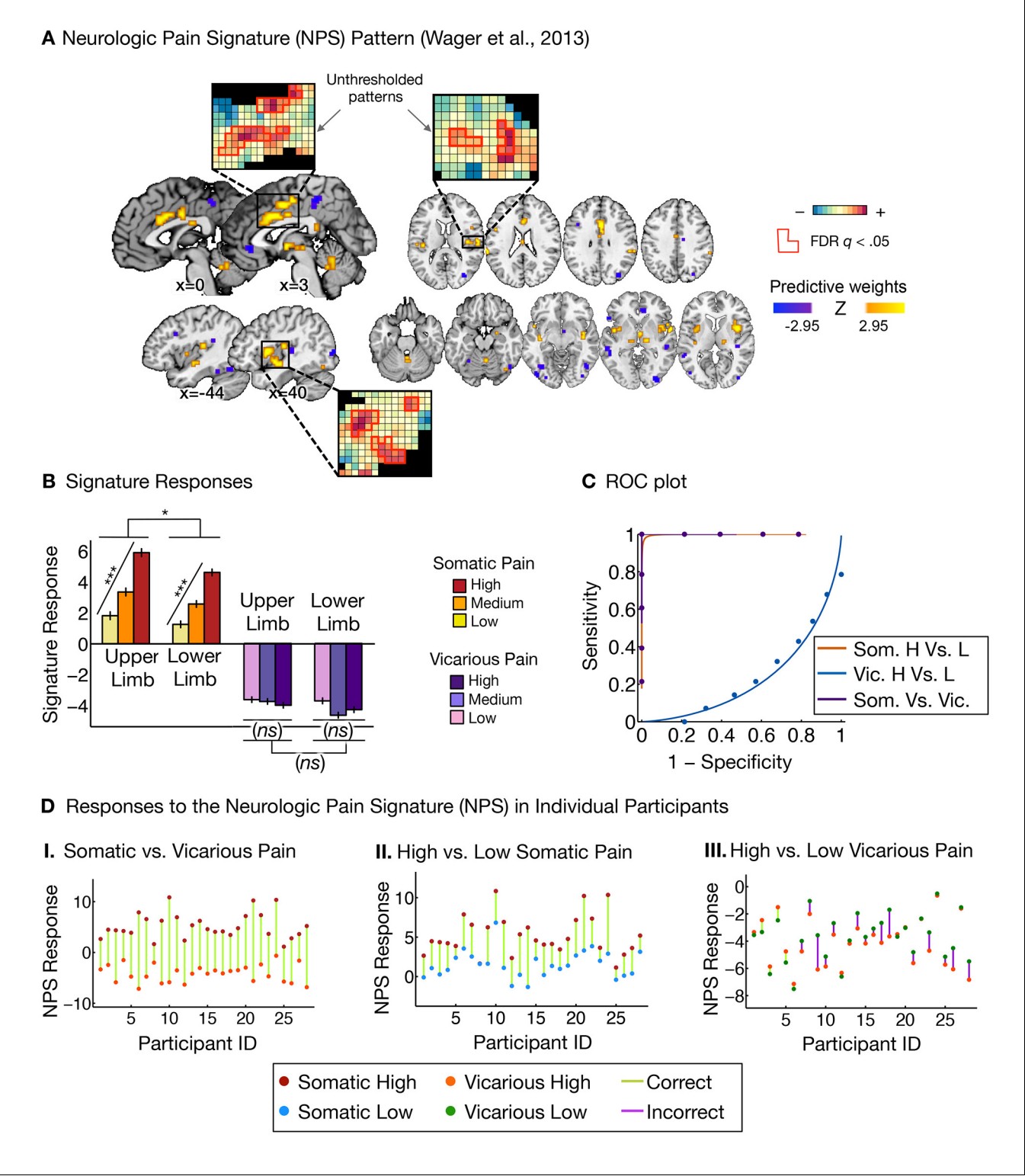

**Figure 2.** Neurologic pain signature (NPS) pattern and analyses. (A) Between-participant thresholded (False Discovery Rate [FDR] q<0.05) Neurologic Pain Signature (NPS) pattern (*Wager et al., 2013*), (all voxels within the NPS were used in the analyses), examples of unthresholded patterns are presented in the insets; small squares indicate voxel weights, black squares indicate empty voxels located outside of the NPS pattern, and red-outlined squares indicate significance at FDR q<0.05; (B) Signature responses computed as the dot product of the NPS pattern weights and estimated activation maps for each participant (including within-participant error bars); (C) Receiver Operating Characteristic (ROC) curves for two-choice forced-alternative accuracies for somatic and vicarious pain, high and low somatic pain, and high and low vicarious pain; (D) Participant-Wise NPS Responses for (I)

*Figure 2 continued on next page*

*Figure 2 continued*

somatic and vicarious pain (accuracy$_{Som-Vic}$ = 100%, p<0.0001), (II) high and low somatic pain (accuracy$_{Hsom-Lsom}$ = 100%, p<0.0001), and (III) high and low vicarious pain (accuracy$_{Hvic-Lvic}$ = 32%, *n.s.*), showing the direction of response for each participant.

Importantly, the NPS response did not increase across levels of vicarious pain on either site ($t_{UL}$(27) = −1.57, *n.s.*, $t_{LL}$(27) = −1.99, *n.s.*, $t_{UL-LL}$(27) = 1.71, *n.s.*; see *Figure 2B and C*). Individual participants also showed the same pattern of NPS responses for both somatic and vicarious pain (*Figure 2D*).

There was a strong de-activation in the NPS response for all vicarious pain conditions, with levels below zero. This decrease was driven by picture-induced activation of regions negatively predictive of pain in the NPS (e.g., ventral occipital cortex, superior temporal sulcus, ventromedial prefrontal cortex; see Appendix and *Appendix 1—figure 1* for additional details).

Additional analyses re-training a new somatic pain pattern on this dataset showed similar results (see Appendix and *Appendix 1—figure 2* for more details) but the NPS is preferred because it was defined *a priori* and its specificity was validated across multiple datasets.

## A novel signature sensitive and specific to vicarious pain

We next sought to identify a distributed pattern of *f*MRI activity that predicts the intensity of vicarious pain experience (i.e., Vicarious Pain Signature or VPS). To parallel the development of the NPS (*Wager et al., 2013*), we used LASSO-PCR (Least Absolute Shrinkage and Selection Operator-regularized Principal Components Regression) to predict the intensity of reported vicarious pain during pain observation (*Wager et al., 2013*). We used a leave-one-participant-out cross-validation (see 'Materials and methods' for details; *Figure 3A* shows the thresholded VPS map using a bootstrap procedure) to get an unbiased test of responses to both somatic and vicarious pain in held-out individuals.

The VPS responded strongly and monotonically to increases in vicarious pain for both upper limb and lower limb sites ($t_{UL}$(27) = 7.42, p<0.0001, accuracy$_{High-Low}$ = 89%, p<0.0001; $t_{LL}$(27) = 10.44, p<0.0001, accuracy$_{High-Low}$ = 100%, p<0.0001; *Figure 3B and C*), with a reduced response for the lower limb site ($t_{UL-LL}$(27)=2.83, p<0.05). Importantly, the VPS showed near-zero responses for all levels of somatic pain, and did not differentiate between somatic pain levels ($t_{UL}$(27) = 1.39, *n.s.*, $t_{LL}$(27) = 0.84, *n.s.*, $t_{UL-LL}$(27) = 0.33, *n.s.*; see *Figure 3B and C*). Individual participants also showed the same pattern of VPS responses for both somatic and vicarious pain (*Figure 3D*).

To test how strongly the VPS's sensitivity and specificity depended on occipital activation—which might be related to enhanced sensory attention—we re-trained the VPS excluding the occipital cortex, with qualitatively identical results (see Appendix and *Appendix 1—figure 3* for more details). The signature response monotonically increased for each level of vicarious pain for both the upper limb and lower limb sites ($t_{UL}$(27) = 5.75, p<0.0001, $t_{LL}$(27) = 5.45, p<0.0001, accuracy$_{Hvic-Lvic}$ = 93%, p<0.0001), but did not differentiate between high and low levels of somatic pain ($t_{UL}$(27) = 1.56, *n.s.*, $t_{LL}$(27) = 1.23, *n.s.*; $t_{UL-LL}$(27) = 0.03, *n.s.*, accuracy$_{Hsom-Lsom}$ = 64%, *n.s.*).

Additionally, we examined whether the observed differences between the NPS and VPS patterns were due to differences in the lateralization of stimuli presentation. As in the studies used to define and validate the NPS, somatic pain was administered on the left limbs. Likewise, as in previous literature, the visual stimuli used to evoke vicarious pain showed injuries to the right limbs. This mirrors the side of painful stimulation from the observer's point of view in an allocentric reference frame, but not an egocentric one; thus, laterality could conceivably play a role. To test this, we repeated the signature response analysis with a left-right flipped version of the VPS pattern (excluding the occipital cortex), which preserves the pattern but with opposite laterality. The results remained the same: The flipped VPS pattern did not track somatic pain intensity ($t_{UL}$(27) = 1.47, *n.s.*, $t_{LL}$(27) = 0.04, *n.s.*, accuracy$_{Hsom-Lsom}$ = 50%, *n.s.*), but did track vicarious pain intensity ($t_{UL}$(27) = 2.68, p<0.05, $t_{LL}$(27) = 2.53, p<0.05, accuracy$_{Hvic-Lvic}$ = 79%, p<0.005). These results indicate that the laterality of the VPS, and by extension the laterality of the stimuli it was trained on, are not an important determinant of its functional properties.

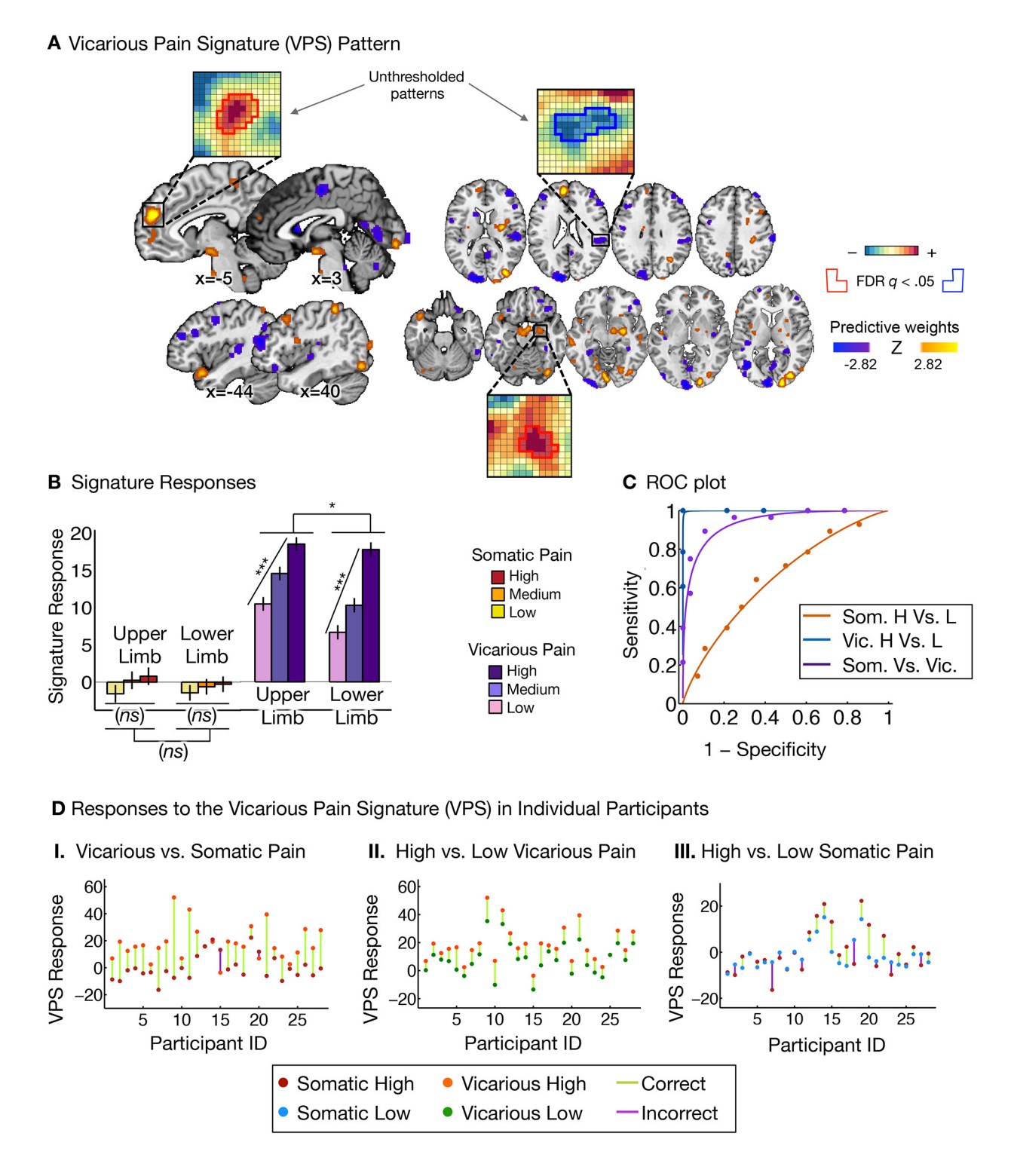

**Figure 3.** Vicarious pain signature (VPS) pattern and analyses. (**A**) Between-participant LASSO-PCR (Least Absolute Shrinkage and Selection Operator-regularized Principal Components Regression) derived pattern for the Vicarious Pain Signature (VPS), and bootstrap thresholded at FDR q<0.05 for display purposes (all voxels within the VPS were used in the analyses), examples of unthresholded patterns are presented in the insets; small squares indicate voxel weights, and red or blue-outlined squares indicate significance at FDR q<0.05; (**B**) Cross-validated signature responses computed from the VPS as the dot product of the VPS pattern weights with the estimated activation maps for each participant (including within-participant standard

*Figure 3 continued on next page*

*Figure 3 continued*

error of the mean); (C) Receiver Operating Characteristic (ROC) curves for two-choice forced-alternative accuracies for vicarious and somatic pain, high and low somatic pain, and high and low vicarious pain; (D) Participant-Wise VPS Responses for (I) vicarious and somatic pain (accuracy$_{Vic-Som}$ = 89%, p<0.0001), (II) high and low vicarious pain (accuracy$_{Hvic-Lvic}$ = 100%, p<0.0001), and (III) high and low somatic pain (accuracy$_{Hsom-Lsom}$ = 64%, *n.s.*), showing the direction of response for each participant.

## Somatic and vicarious pain: encoded in dissociable brain systems

As described above, each of the two patterns we identified (the NPS and VPS) was influenced by only one type of 'pain' induction: Somatic pain induced intensity-dependent responses in the NPS only, and vicarious pain induced intensity-dependent responses in the VPS only. This pattern of results shows *separate modifiability* of these patterns, a strong inferential criterion by which two processes are functionally independent if experimental manipulations can modify each pattern without affecting the other one (*Sternberg, 2001*). It rules out common influences of general shared processes such as increased allocation of attention [e.g., (*Woo et al., 2014*)].

In addition, the two patterns were anatomically distinct. Regions in the NPS that most reliably predicted somatic pain included aINS, dACC, dorso-posterior insula (dpINS), and SII. By contrast, VPS regions encoding vicarious pain included the dmPFC, amygdala, PCC, and TPJ. Moreover, the predictive weights in the NPS and VPS were orthogonal, showing near-zero spatial correlations across the brain ($r = -0.03$; 104,360 voxels in the NPS & VPS) and within key regions (see below).

Additional analyses showed that both the NPS and VPS were activated in response to the somatic or vicarious pain stimulus specifically during the stimulation period, with the expected time course. Both the NPS and VPS responded only during somatic and vicarious pain events, respectively, with no anticipatory activity (*Figure 4A and B*). There were small differences in the time courses of responses to somatic and vicarious pain, which are expected because somatic pain summates across time and peaks after stimulus offset (*Koyama et al., 2004*), whereas vicarious pain does not have this physiological property. Critically, we found no evidence for 'off-target' responses (VPS responses to somatic pain or NPS responses to vicarious pain) at any point during the trial.

Another issue that may make the NPS and VPS responses appear more dissimilar is differential habituation or sensitization across trials. To test this, we estimated NPS and VPS amplitudes for each trial using a 'beta series' approach (*Mumford et al., 2012*) and examined their stability across trials. We did not see any evidence for systematic variation [e.g., sensitization or habituation; (*Jepma and Wager, 2013*)] across trials (see Appendix and *Appendix 1—figure 4*).

Additionally, we tested whether the VPS pattern was similar to whole-brain patterns we identified as related to (a) romantic rejection (*Woo et al., 2014*) and (b) negative emotion induced by affective pictures (*Chang et al., 2015*), two other types of aversive experience. The VPS was uncorrelated with either pattern ($r = -0.03$ and $r = 0.03$, respectively), demonstrating that the signature for vicarious pain is not a marker for general negative emotion, unlike previous results (*Corradi-Dell'Acqua et al., 2011*), and supporting the specificity of the VPS for pain empathy.

## Dissociable local representations for 'pain affect'

Theories of pain empathy have focused on the dACC and aINS as critical for shared representations of pain affect, and emphasize overlapping activity in spite of differences in the sensory system involved (e.g., somatic vs. visual). Like previous work (*Corradi-Dell'Acqua et al., 2011*; *Lamm et al., 2011*; *Singer et al., 2004*), we found strong activation in these regions in both somatic and vicarious pain in standard univariate analyses (*Figure 5A[I] and A[III]*). We identified regions of overlapping activation in the dACC and aINS (*Figure 5B*), and tested the similarity and separate modifiability of the pain-predictive patterns only within these 'shared' regions.

The NPS and VPS pattern weights within the aINS and dACC were negatively correlated (aINS: $r = -0.31$ [349 Voxels]; dACC: $r = -0.41$ [1,125 Voxels]; see *Figure 5C*), indicating a lack of positive shared representation. Additionally, we used data from each region of interest (ROI) to train local patterns predictive of somatic and vicarious pain intensity within individual participants. These patterns were also spatially uncorrelated within each region (aINS: mean $r = 0.0185 \pm 0.0092$, *n.s.*; dACC: mean $r = 0.0350 \pm 0.0195$, *n.s.*). Further analyses indicated that both dACC and aINS successfully tracked somatic pain (cross-validated accuracy$_{High-Low}$ for aINS: 89%, p<0.0001; dACC:

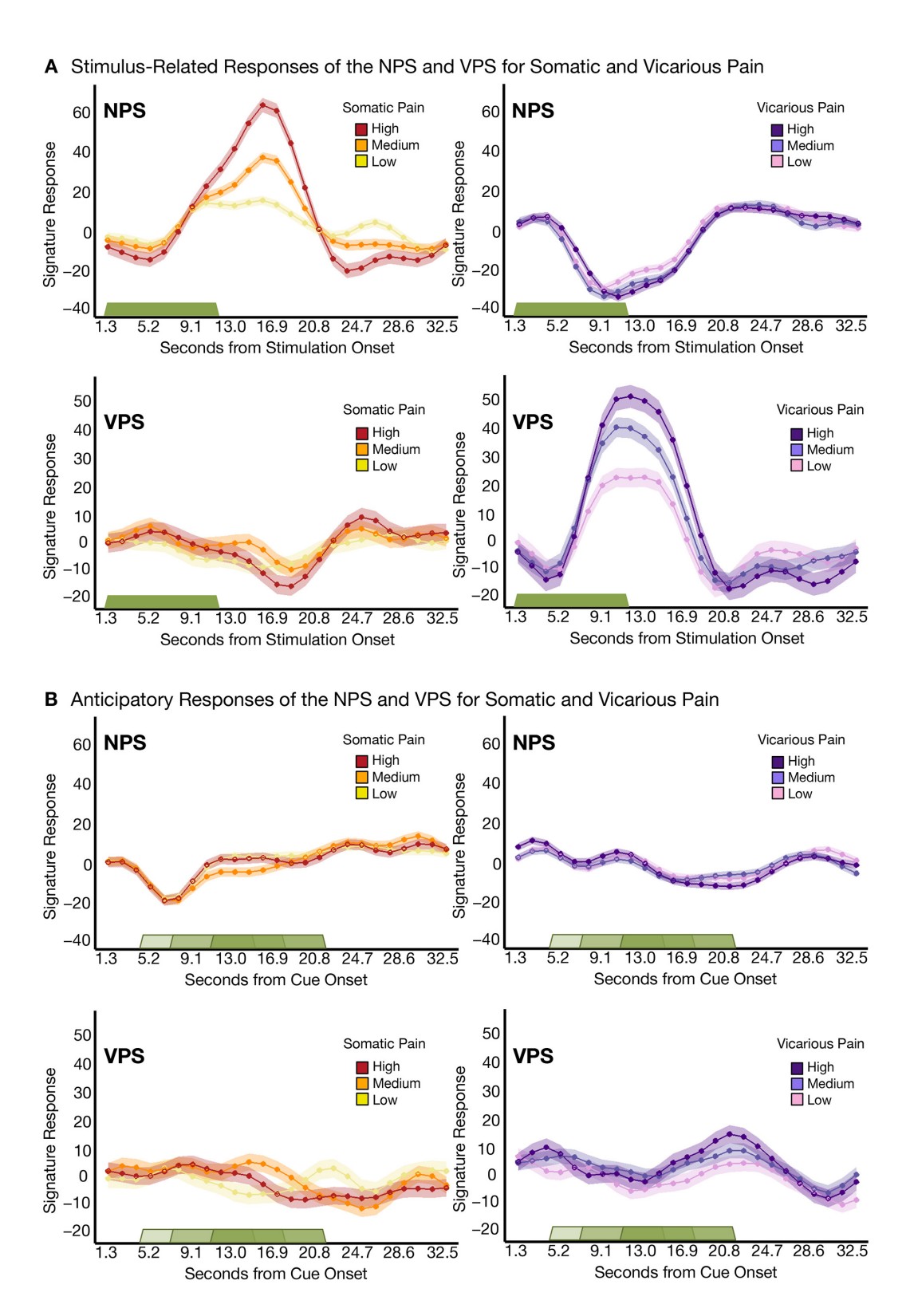

**Figure 4.** Trial-Level finite-impulse response (FIR) analyses. (**A**) Stimulus-related activity for the NPS and VPS for somatic and vicarious pain (including within-participant standard error of the mean (shading), where the green bar shows stimulus duration; (**B**) Anticipatory (cue-locked) activity for the NPS

*Figure 4 continued on next page*

*Figure 4 continued*

and VPS for somatic and vicarious pain (including within-participant error (shading), where the green bars show stimulus duration after a jittered pre-stimulus anticipatory fixation.

75%, p<0.01). Vicarious pain, however, showed marginal within-modality classification (aINS: 71%, p<0.05; dACC: 57%, *n.s.*), suggesting that local patterns may be insufficient to represent vicarious pain. Importantly, cross-prediction analyses showed that the somatic pain pattern did not predict vicarious pain intensity in both aINS and dACC, and vice versa (all accuracy values < 54%, *n.s.*), demonstrating separate modifiability. Together, these results support the independence of somatic and vicarious pain representations in the aINS and dACC. In addition, they suggest that vicarious pain representation is distributed across regions, and can best be captured in whole-brain but not local analyses.

Finally, we used within-individual searchlight analysis (8 mm sphere) to test (a) whether any local region of the brain was highly predictive of somatic or vicarious pain, and (b) whether there was strong shared representation in any local region (*Kriegeskorte et al., 2006*). The within-modality patterns exhibited a similar spatial topography to our whole-brain analyses (see *Figure 6A–D*), but no results for vicarious pain survived false discovery rate (FDR) correction for multiple testing. Importantly, the effect sizes even for the most predictive regions were an order of magnitude smaller than those observed in our whole brain analyses: The maximal within-modality effect size out of 181,129 local regions tested was $r^2 = 0.06$ for somatic pain and $r^2 = 0.018$ for vicarious pain. In addition, no cross-prediction results survived FDR correction, and the distribution of cross-prediction results across regions was centered on zero for both somatic and vicarious pain (see *Figure 6A–D*). Again, effect sizes across the brain were small: The largest effects were $r^2 = 0.011$ (somatic to vicarious) and $r^2 = 0.017$ (vicarious to somatic), indicating that local regions are insufficient by themselves to accurately predict ratings, and distributed patterns are required. Together, these results suggest that though there may be weak shared pattern information (*Corradi-Dell'Acqua et al., 2011*; *2016*), local patterns in the aINS and dACC are insufficient to serve as representations for vicarious pain. In addition, separate cross-validated prediction analyses within-individuals revealed that whole brain spatial patterns predictive of somatic and vicarious pain were spatially uncorrelated (mean $r = 0.0055 \pm 0.0065$ S.E., *n.s.*; see Appendix and *Appendix 1—figure 5*).

## Somatic pain and vicarious pain have divergent somatotopy

The results presented above demonstrate that both the NPS and VPS generalize across upper and lower limb sites. However, our experimental design also allowed us to identify body-site specific representations and provide a preliminary comparison of the somatotopic organization of somatic and vicarious pain. Shared somatotopic organization (e.g., if vicarious pain on the upper limb activated upper limb-specific somatosensory regions) would provide evidence for shared representation, while divergent somatotopy would provide further evidence for differential brain representation.

We trained between-participant support vector machine (SVM) classifiers to differentiate between upper and lower limb sites separately for somatic and vicarious pain (we also performed within-participant SVM classification for completeness, with the same results; see Appendix for details). We then compared the body site-predictive maps across somatic and vicarious pain. The leave-one-participant-out SVM classifier successfully discriminated somatic pain on upper limb (UL) versus lower limb (LL) sites (see *Figure 7A*; $t_{Som.Pain}(27) = 6.90$, p<0.0001; accuracy$_{UL-LL}$= 93%, p<0.0001). The regions that made reliable contributions to classification (as tested with a bootstrap procedure; see 'Materials and methods' for details) paralleled the somatotopy identified in previous literature, including specific regions of contralateral SI, mid INS, and dpINS (*Baumgärtner et al., 2010*; *Brooks et al., 2005*; *Hua et al., 2005*; *Picard and Strick, 1996*). For vicarious pain, SVM classification also discriminated UL versus LL (see *Figure 7A*; $t_{Vic.Pain}(27) = 6.12$, p<0.0001; accuracy$_{UL-LL}$= 93%, p<0.0001; within-participant average accuracy$_{UL-LL}$= 100%, p<0.0001). Crucially, the somatotopic patterns for vicarious pain did not include the expected topography in contralateral SI, SII, mid-INS or dpINS, in either hemisphere (*Figure 7B and C*). Instead, vicarious pain somatotopy was represented in other brain areas, including the supplementary motor area, anterior cingulate (ACC),

**A** General Linear Model (GLM) Analyses for Somatic and Vicarious Pain

**I.** Somatic pain vs. baseline (FDR q < .01)

**II.** Vicarious pain vs. baseline (FDR q < .01)

**B** Overlapping Regions from the GLM Analyses for Somatic and Vicarious Pain

Somatic pain ($q < .01$)

Overlap

Vicarious pain ($q < .01$)

**C** Pattern Similarity between NPS and VPS within GLM Overlaps

**dACC**

NPS     VPS

$r = -0.41$

-5     0     5     x 10⁻³

right **aINS**
(for display)

NPS     VPS

$r = -0.31$
(bilateral)

-5     0     5     x 10⁻³

**Figure 5.** Univariate general linear model analyses and multivariate pattern comparisons. (**A**) group level general linear model (GLM) analyses for somatic and vicarious pain. (I.): GLM results for somatic pain against baseline thresholded at FDR q<0.01; (II.): GLM results for vicarious pain against baseline thresholded at FDR q<0.01; (**B**) General Linear Model (GLM) results for somatic pain (in orange) and vicarious pain (in purple) against baseline, thresholded at FDR q<0.01 with overlap in the statistically significant regions (FDR q<0.01) between somatic and vicarious pain shown in yellow (these overlapping regions were used for further within-participant cross-prediction analyses); (**C**) Pattern comparison within the anterior insula (aINS) and dorsal anterior cingulate cortex (dACC) for the NPS and VPS (black squares in the pattern mask indicate empty voxels not included for the analysis); correlation values are computed using all overlapping voxels in the dACC and bilateral aINS voxels (only right aINS shown for display).

and medial prefrontal (mPFC) cortices (see Appendix and *Appendix 1—figure 6* for additional analyses).

Thus, somatotopy was apparent for both somatic and vicarious pain and strongly predictive of upper versus lower limb stimulation, but the fMRI patterns were qualitatively distinct in each

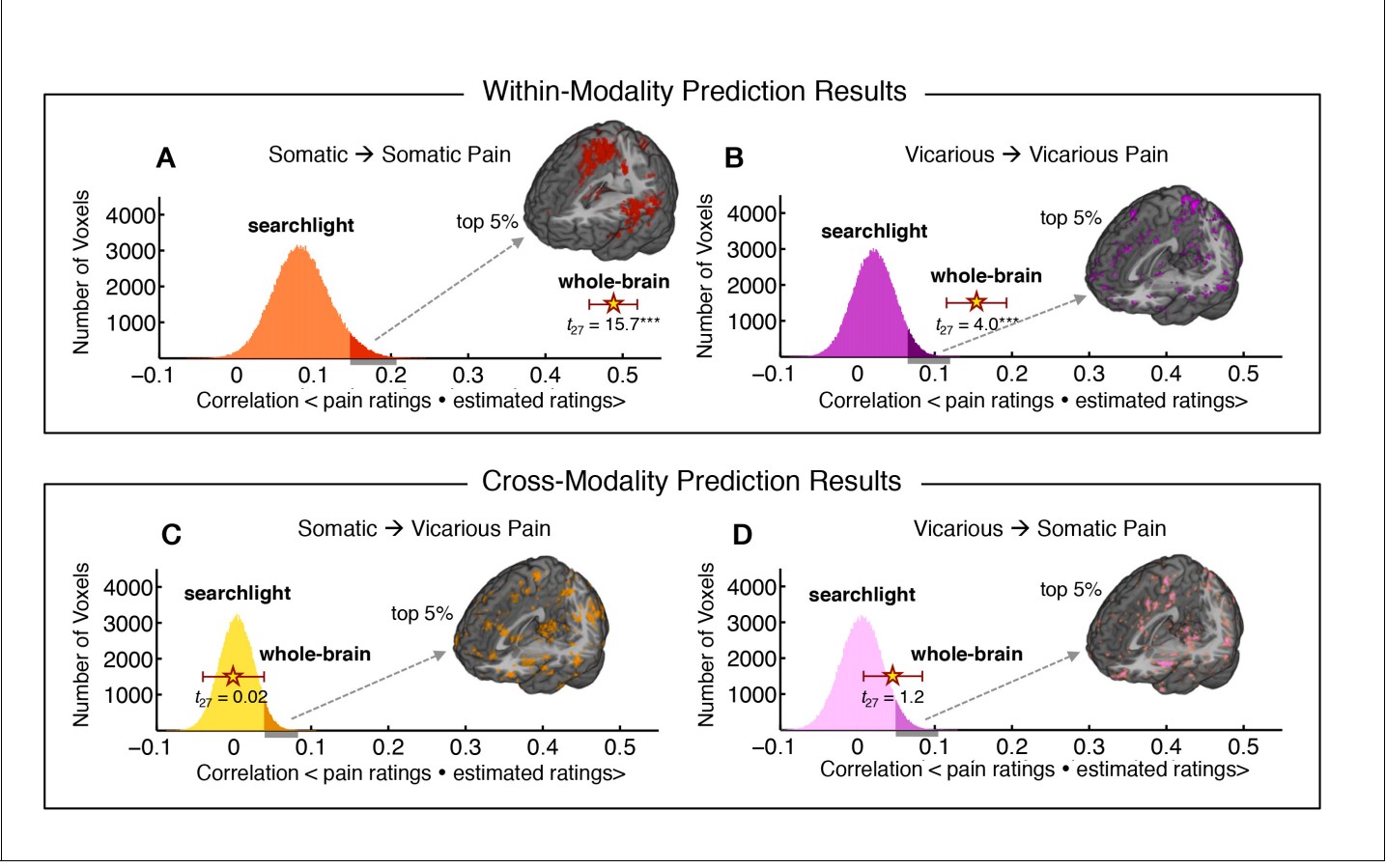

**Figure 6.** Within-participant whole-brain and local searchlight analyses. (**A—D**) Distribution of correlations between actual behavioral ratings and estimated behavioral ratings for within and cross-modality predictions from whole-brain and local searchlight analyses. **Histograms**: Mean outcome correlation from local searchlight analysis computed across participants (dark shades represent voxels with top 5% of correlation values); **Stars with Error bar**: Mean outcome correlation computed across participants from whole-brain analysis (error bars represent standard error of the mean [SEM]); **A** and **B** Whole brain and local searchlight results from cross-validated *within-modality* (somatic to somatic; vicarious to vicarious) predictions. Brain maps show voxels with the top 5% correlations; **C** and **D** Whole brain and local searchlight results from *cross-modality* (somatic to vicarious; vicarious to somatic) prediction with brain map showing voxels with the top 5% correlations.

condition. Somatotopy is the primary way of identifying somatosensory cortical representations (*Penfield and Rasmussen, 1950*); thus, our preliminary finding of the expected somatotopy for somatic pain but not vicarious pain suggests that vicarious pain does not involve re-activation of somatosensory representations. Rather, body site-specific representations of vicarious pain may be accomplished using the mPFC 'mentalizing' system and perhaps other ideomotor systems.

## Replication and generalization (Study 2 and Study 3)

To test the generalizability of the NPS across different types of noxious input, we analyzed data from two additional *f*MRI studies (Study 2 and Study 3) that used other types of somatic pain. Study 2 (N = 28) used mechanical pressure pain at two intensities—4 kg/cm$^2$ and 6 kg/cm$^2$—applied on the right thumbnail. The NPS responded more strongly to high versus low pressure, demonstrating sensitivity to mechanical pain ($t_{High-Low}$(27) = 3.12, p<0.005; accuracy$_{High-Low}$ = 71%, p<0.05; see *Figure 8A*). The discrimination accuracy is limited here by the inclusion of relatively few trials (5 per condition). Nevertheless, these results show that the NPS successfully predicts intensity in the somatic pain regardless of the site of the stimulation or type of somatic pain.

Study 3 (N = 15) used electrodermal (electrical or shock) pain delivered to the left ankle, and also included an observed pain condition with visual stimuli depicting pain on upper and lower limbs. However, perspective-taking was not employed in this study—the instructions emphasized

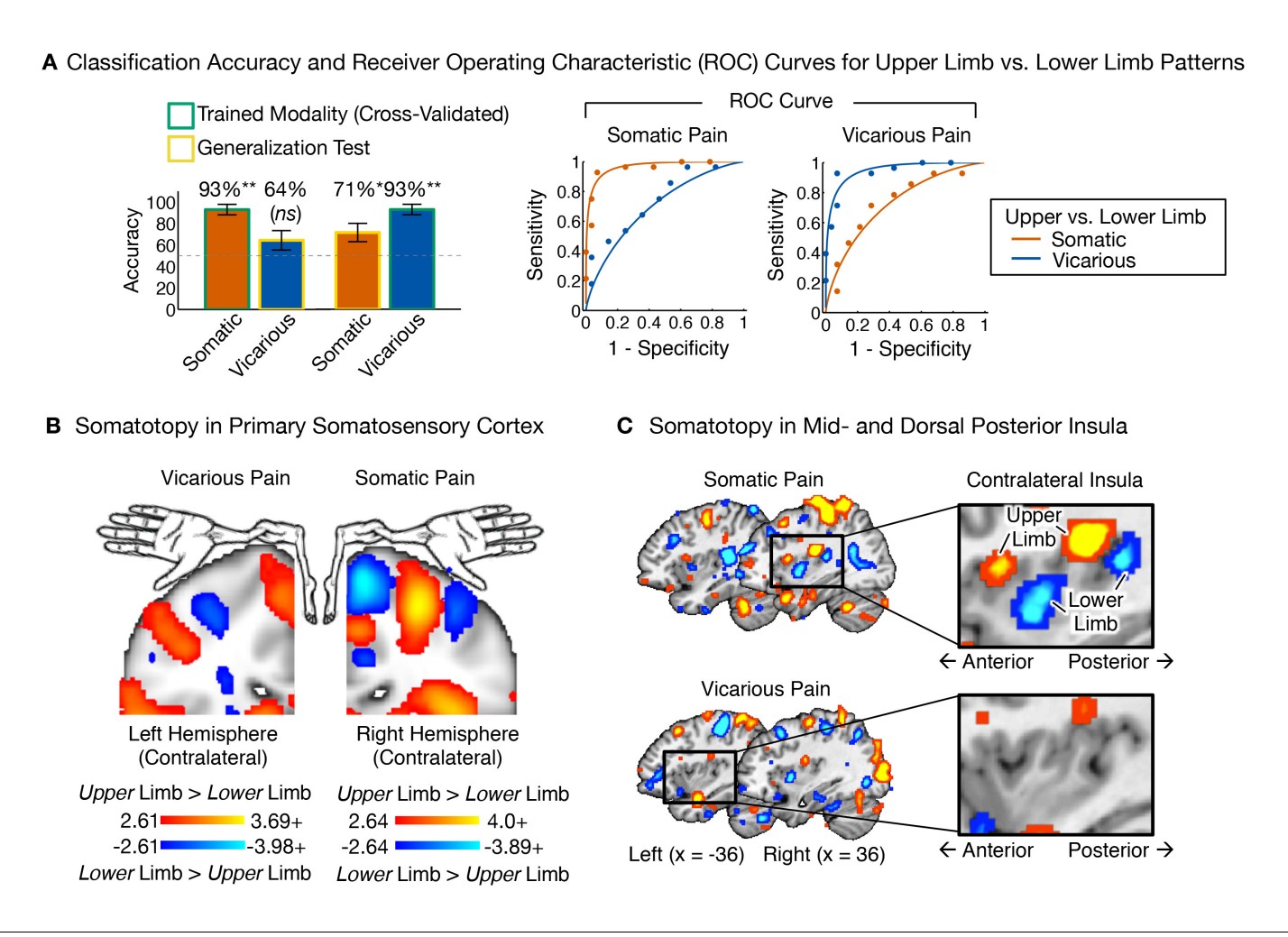

**Figure 7.** Upper limb versus lower limb multivariate patterns for somatic and vicarious pain. (**A**) Accuracy statistics for upper limb versus lower limb weight maps for somatic and vicarious pain; Receiver Operating Characteristic (ROC) curves for two-choice forced alternative tests for upper versus lower limb sites; (**B**) Support Vector Machine (SVM) derived pattern for upper limb versus lower limb sites for somatic and vicarious pain, bootstrap thresholded at FDR q<0.05 for display purposes, showing somatotopy for these sites in the primary somatosensory cortex (schematic of homunculus shown for reference); (**C**) SVM derived pattern for upper limb versus lower limb sites for somatic and vicarious pain, bootstrap thresholded at FDR q<0.05 for display purposes, showing somatotopy for these sites in the mid- and dorsal-posterior insular cortex (warm colors indicate upper limb regions and cool colors indicate lower limb sites).

observation only rather than engage in perspective-taking—so VPS activation was expected to be weaker than in Study 1. The NPS responded more to shock pain than observed pain ($t_{Shock-Obs}(14)$ = 5.58, p<0.0001; accuracy$_{Shock-Obs}$ = 100%, p<0.0001; see *Figure 8B*, left). Conversely, the VPS responded to more to observed pain than shock pain ($t_{Obs-Shock}(14)$ = 3.05, p<0.005; accuracy$_{Obs-Shock}$ = 73%, *n.s.*; see *Figure 8B*, right). Thus, the findings from Study 3 show that the NPS and VPS dissociate somatic versus observed pain in an independent sample.

## Discussion

The ability to vicariously experience others' pain is critical for empathy and prosocial behavior. Neuroimaging can help understand *how* we represent others' suffering by testing whether vicarious pain activates the same neural mechanisms as somatic pain, implying shared affective experience. Overlapping neural responses in the dACC and aINS (*Hutchison et al., 1999*; *Jackson et al., 2006b*;

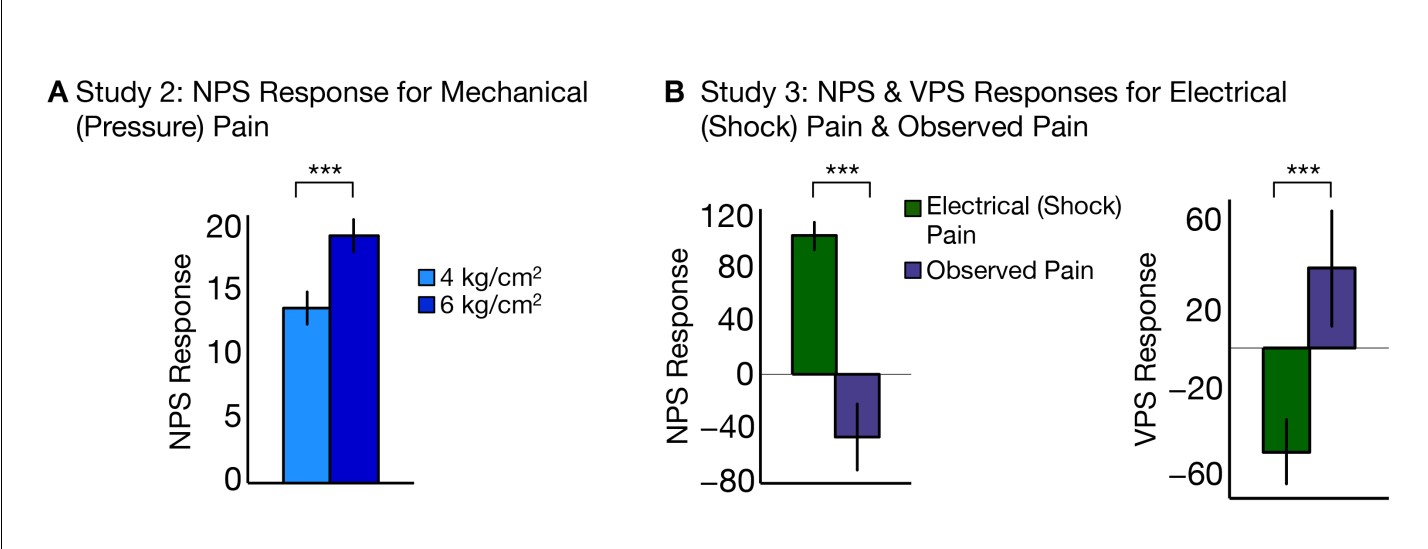

**Figure 8.** NPS and VPS responses for Study 2 and Study 3. (**A**) Study 2 – NPS responses for mechanical (pressure pain) delivered at low and high intensities, (i.e., 4 kg/cm² and 6 kg/cm²) computed as the dot product of the NPS pattern weights and estimated activation maps for each participant (including within-participant standard error of the mean; total of five trials per level of stimulation per participant; (**B**) Study 3 – NPS (right) and VPS (left) responses for electrodermal (electrical or shock) pain and observed pain, computed as the dot product of the respective pattern weights for NPS (right) and VPS (left) and estimated activation maps for each participant (including within-participant standard error of the mean.

*Lamm et al., 2011*; *Singer et al., 2004*) have been interpreted in terms of shared pain representations, but these responses may reflect emotions that are distinct from pain, and need not entail shared experience (*Hooker et al., 2008*; *Zaki, 2014*). The question of shared representation can only be addressed if brain markers that are sensitive, specific, and generalizable can be identified for both somatic and vicarious pain. Then, the similarity of those markers can be assessed. But can a representation of vicarious pain be identified, and if so, is it conserved across individuals so that it can be prospectively compared with somatic pain and related to empathetic behavior? Our findings suggest that the answer to this question is yes. Is this representation similar to that for somatic pain? Our findings suggest that the answer to this question is no.

In a series of analyses across 3 studies, we identified distinct fMRI patterns that encoded the intensity of somatic and vicarious experiences, suggesting that somatic and vicarious pain are distinct processes. Local brain patterns did not accurately predict somatic or vicarious pain experience. Instead, distributed patterns within and across multiple association regions strongly and separately predicted both somatic and vicarious pain. Somatic pain was encoded largely in the aINS, dACC, dpINS, and SII—regions targeted by ascending nociceptive afferents (*Bushnell et al., 2013*; *Willis, 1985*) and thought to encode sensory and affective aspects of pain. By contrast, vicarious pain was encoded in portions of the dmPFC, amygdala, PCC, and TPJ—multisensory, heteromodal regions associated with mentalizing (*Hampton et al., 2008*; *Mitchell et al., 2006*; *Saxe et al., 2003*), consistent with other work emphasizing the importance of mentalizing systems in empathy (*Hooker et al., 2008*; *Rütgen et al., 2015*; *Zaki, 2014*). These findings demonstrate separate modifiability (*Sternberg, 2001*; *Woo et al., 2014*), which entails sensitivity and specificity of each brain pattern to only one type of 'pain', and rules out common influences of shared demands on attention, 'salience,' 'arousal,' and other more general processes (see Appendix for an extended discussion).

In addition, strong somatotopy was found for both somatic and vicarious pain, though the somatotopic organization was very different for each. Somatic pain and touch have a well-established somatotopy in SI, SII, and dpINS (*Baumgärtner et al., 2010*; *Brooks et al., 2005*; *Hua et al., 2005*), which we observed here. Vicarious pain showed different somatotopic patterns that, interestingly, were localized within 'mentalizing' systems rather than somatosensory ones.

## A departure from shared representation

Most previous studies of vicarious pain point out its similarity with somatic pain, and it is thus widely believed that the two experiences rely on the same systems. Why are our findings and conclusions different? There are three main reasons.

First, many previous studies focused only on the points of similarity—mainly identified in two isolated brain regions, dACC and aINS—ignoring dissimilarities [but cf. (*Lamm et al., 2011*), who write, 'our results reveal more differences than similarities...' p. 2500]. Here, we aimed for an unbiased assessment of the two processes.

Second, most previous work has identified 'pain-related' activation by contrasting pain with loosely matched control conditions (e.g., neutral or innocuous stimuli). Overlapping activity in such contrasts may be caused by many processes that are not pain, including general negative affect, attention, and arousal [see (*Wager et al., 2016*) for discussion]. Engagement of these processes may be responsible for the similar activation in previous work. By contrast, in our study, we attempted to isolate pain-relevant patterns that predicted the magnitude of experienced intensity, and examined the similarity of those patterns.

Third, most previous studies focused on voxel-wise activation 'blobs,' rather than on multivariate patterns, which can be sensitive to information at finer spatial scales (*Shmuel et al., 2010*), possibly even below the intrinsic resolution determined by the voxel size (*Kamitani and Tong, 2005*). This property, combined with our experimental approach targeting within-person variations in pain intensity, suggests that the multivariate patterns we identified are more likely to reflect specific representations contained in meso-scale neural circuits.

Based on our findings, we infer that the overlapping activation in the dACC, aINS, and other areas is not related to shared pain experience. Interestingly, on close reading, the few previous multivariate pattern-based studies agree broadly with this interpretation. The brain patterns they identified as shared across somatic and vicarious pain were not specific to 'pain,' as these patterns were also activated by other, non-painful types of negative affect (*Corradi-Dell'Acqua et al., 2016*; *Zaki et al., 2016*). As in our study, in these studies what is shared does not seem to be particular to pain per se.

## Identifying representations: criteria and evidence

Somatic and vicarious pain experiences, *by definition*, are similar in some ways and different in others. They are similar in that they are both aversive, salient, and attention-capturing. They differ in their sensory input modalities (somatosensory versus visual) and likely many of the cognitive processes involved. Brain activity, including both regional activation and multivariate patterns, may reflect any or all of these processes. For this reason, it is necessary to test the similarity of the brain patterns that *most closely encode pain experiences of each type*. We have proposed three criteria for identifying patterns that may be used to make inferences about pain representation. Such patterns should be (a) sensitive to pain, and thus respond in proportion to its experience; (b) specific to pain, and not activated by non-painful events; and (c) generalizable across varied instances of the type of pain. For example, a brain pattern that encodes 'pain affect'—whether common to self- and other-pain or not—should increase as pain affect increases, and not respond to non-painful forms of negative emotion such as disgust (*Zaki et al., 2016*). Previous studies have identified both shared and dissimilar patterns of activity (*Corradi-Dell'Acqua et al., 2011*; *2016*) in local dACC and aINS regions. Though an important step, these patterns have not been shown to track pain experience with meaningfully large effect sizes, and our findings here suggest that local regions are insufficient to do so. In addition, they have not identified a pattern that tracks 'pain affect' specifically and does not respond to other negative emotions.

Our findings provide an advance because they identify patterns that are strongly predictive of pain experience. Though they do not test many other kinds of emotions, the patterns are demonstrably specific to either pain for oneself or another. In addition, a lack of correlations between the NPS, VPS, and patterns predictive of rejection and negative emotion further suggest that these patterns are not sensitive to general emotional arousal. Fortunately, the between-participant patterns we identified here can be prospectively tested for generalizability to other emotions and paradigms in future studies. Indeed, in an initial step towards broader generalizability, we showed that the

sensitivity and specificity of both patterns generalize to new studies, and that our putative somatic pain representation, the NPS, generalizes across three types of somatic pain.

The fact that capturing vicarious pain experience requires identifying distributed patterns across large-scale networks is consistent with prior work suggesting that accurate predictions of pain and emotional experiences requires combining signals across brain regions and networks (*Brodersen et al., 2012*; *Cecchi et al., 2012*; *Chang et al., 2015*; *Kassam et al., 2013*; *Kragel and Labar, 2013*; *Nummenmaa et al., 2014*; *Wager et al., 2015*). This work fits in with evidence for *population coding* in motor processing (*Georgopoulos et al., 1992*), spatial processing (*Fried et al., 1997*), and other domains (*Davis and Poldrack, 2013*; *Haxby et al., 2014*; *Panzeri et al., 2015*). It also converges with views of somatic pain as represented by patterns across populations of neurons distributed across macroscopic brain regions (*Coghill et al., 1999*; *Leknes and Tracey, 2008*; *Melzack and Wall, 1967*). Thus, it may be useful to explicitly consider population coding hypotheses in future studies, at both the *f*MRI and neural levels of analyses.

## Limitations and future directions

Our approach is not biased towards or against finding common or distinct representations. It is grounded in widely accepted approaches to testing cross-stimulus type prediction (*Bruneau et al., 2013*; *Etzel et al., 2008*; *Kourtzi and Kanwisher, 2001*; *Parkinson et al., 2011*). However, our approach is not guaranteed to be maximally sensitive to shared representations. Optimizing for shared representation would require using machine-learning approaches to predict pain in a modality-nonspecific fashion. Another important issue is individual differences in the experience of vicarious pain. Having established pain-predictive brain patterns as we do here, subsequent studies could include stimuli that are externally matched, and brain responses in the 'vicarious pain signature' are correlated meaningfully with empathy ratings across participants. In addition, the VPS could potentially be used to test clinical populations: for example, whether patients in chronic pain show greater (or lesser) responses to others' suffering. These questions are beyond the scope of the present study, but could be undertaken in future work.

Our study used conditions designed to maximize the neural overlap between vicarious and somatic pain, by using 'painful' pictures previously demonstrated to induce stronger somatosensory activity—and are thus potentially *more* somatic pain-like—than other, 'cued empathy' paradigms (*Lamm et al., 2011*). In addition, we instructed participants to adopt a perspective-taking stance previously shown to enhance activation in dACC, aINS, and somatosensory cortex (*Jackson et al., 2006a*). Further studies are needed to assess the impact of instructions, mental stance, and which aspects of pain (e.g., sensory, affective) are being evaluated on the activation of the VPS (*Zaki et al., 2016*), and to assess generalizability across different types of vicarious pain paradigms.

## Conclusions

This work provides a specific target signature, the Vicarious Pain Signature (VPS), which can be used prospectively for empathy-related activity in future studies. The VPS may help us understand factors that promote or impair vicarious pain experience, and its impact on prosocial behavior (*Craig et al., 2010*).

Perceiving others' pain does not appear to recruit the same neural circuitry as experiencing the pain ourselves. Rather than recruiting our somatosensory system to understand another's pain, we use processes involved in representing another's mental state. The lack of direct representation of others' pain in somatic pain systems provides a mechanism for understanding why we might systematically under-weigh others' painful experiences, including their suffering (*Gilbert et al., 1998*; *Loewenstein, 1996*; *Van Boven and Loewenstein, 2003*), and substantiates Adam Smith's insight from 250 years ago that our moral sentiments are grounded in our cognitive rather than sensory faculties.

## Materials and methods

Data from three different studies were used in this project. Study 1 was the primary study that was used to develop the methodology discussed in the paper. Study 2 and Study 3 were part of separate projects that involved clinical populations, but only data from the healthy normal controls were used to validate the methodology from Study 1.

Study 1 (N = 28) is part of a large ongoing project examining cue-based expectation effects, and predictive errors for mismatched cue validity across multiple aversive modalities. However, here we focused on machine-learning based prediction of intensity of somatic and vicarious pain, and the predictive cue effects were designed to be orthogonal to the intensity of stimulation. Study 1 served as a training set for the vicarious pain signature (VPS), a somatic pain predictive pattern, and was also used to identify within-participant somatic and vicarious pain patterns. Study 1 additionally served as a test set for the Neurologic Pain Signature (NPS).

Study 2 (N = 28) is part of a study on clinical pain sensitivity differences between healthy normal controls and patients with fibromyalgia [see (*López-Solà et al., 2010a*; *2010b*; *2014*) for related published work]. Here, Study 2 served as a test set for the NPS to examine whether the NPS was sensitive and specific to a different type of somatic pain (i.e., mechanical pressure pain), and whether the pattern generalized to different body locations (i.e., the right thumbnail).

Study 3 (N = 15) is part of a study comparing the cognitive and affective processes for somatic and observed pain in healthy normal controls and patients diagnosed with Autism Spectrum Disorder. Here, Study 3 served as a test set for both the NPS and VPS to examine whether these patterns dissociated somatic from observed pain conditions. In addition, Study 3 involved electrodermal (electrical or shock) stimulation on the left ankle, which served as another test of the generalizability of the NPS across somatic pain types and body sites.

## Study 1

### Participants

Thirty healthy, right-handed participants (Mean Age = 25.2 years, SD = 7.4 years, 12 Females) were recruited for a multiple session (i.e., somatic and vicarious pain) functional magnetic resonance imaging (fMRI) experiment. Participants completed an additional session on aversive (bitter) taste, which will be described in a separate manuscript. Two female participants did not complete the somatic pain session so the final sample size was 28. The order of sessions was counterbalanced, and there were no significant order effects between participants. All participants provided informed consent, and all experimental procedures were conducted with the approval of the Institutional Review Board of the University of Colorado Boulder. Preliminary eligibility of participants was determined through an online questionnaire, a pain safety screening form, and an fMRI safety screening form. Participants with psychiatric, physiological or pain disorders, and neurological conditions were excluded.

### Pain calibration

All participants completed a pain calibration session to determine if they could tolerate the somatic stimulations that they would receive in the actual fMRI experiment. Somatic stimulation was administered as thermal pain, which was applied on the volar surface of the left forearm (i.e., upper limb) and dorsal surface of the left foot (i.e., lower limb) using a TSA-II Neurosensory Analyzer (Medoc Ltd., Chapel Hill, NC) with a 16 mm Peltier thermode end plate. Three levels of somatic stimulation (pseudo-randomly assigned to each participant)—low (44 or 45°C), medium (46 or 47°C) and high (48 or 49°C)—were applied to four different locations on both the upper limb and lower limb . Each stimulation lasted a total of 11s with a 2s ramp-up and a 2s ramp down, and 7s at the peak target temperature. The participants made responses on a Visual Analog Scale (VAS), which had anchors based on a labeled magnitude rating scale (*Bartoshuk, 2000*; *Green et al., 1996*) for No Sensation (0% of scale length), Barely Detectable (1.4% of scale length), Weak (6.1% of scale length), Moderate (17.2% of scale length), Strong (35.4% of scale length), Very Strong (53.3% of scale length), and Strongest Imaginable Sensation (100% of scale length). Participants first made a moment-by-moment rating where for the duration of the trial, as they experienced any sensation, they used a pointer on the screen to move continuously along the rating scale and indicate the level of sensation they felt at each moment. They then made an overall rating at the end of each trial to indicate the maximum overall sensation they experienced in that trial. As part of the calibration procedure, participants also viewed pictures for 11s (to match the thermal pain trials) depicting varying levels of injury occurring on the right hand or foot (*Jackson et al., 2005*). Participants were given the instruction, 'Imagine the experience if the situation in each picture happened to you, and rate how much sensation you would feel in that situation'. They used the same visual analog scale as the somatic pain calibration to make their ratings—a moment-by-moment rating while they viewed the picture

and an overall rating about their experience after each trial. Participants who successfully completed the calibration procedure were then scheduled for the fMRI experimental sessions.

## Experimental sessions

All imaging data were acquired on a Siemens Tim Trio 3T MRI scanner in the Intermountain Neuro-imaging Consortium facility at the University of Colorado Boulder.

Participants completed two separate counterbalanced scanning sessions, which were otherwise identical except that they were administered different types of stimulation for each session (i.e., somatic or vicarious pain). Each scanning session contained 11 runs and lasted about an hour. The experimental design was the same for both sessions, with the only difference being the duration of intra- and inter- stimulation jitter. Each stimulation was preceded by one of three levels of predictive cues that corresponded to three levels of stimulation, with the cues and stimulation completely crossed with each other (i.e., a 3 × 3 experimental design). These predictive cues were designed to be orthogonal to the intensity of stimulation, and were used to study cue-based expectation effects for aversive experiences (to be addressed in a separate manuscript). Prior to being scanned, participants completed a short training with an explicit learning task in which they learned the levels of the three cues that were to be later presented in the scanner (*Atlas et al., 2010*). The first two fMRI runs consisted of a conditioning task where the participants learned the association between the cues they encountered in the pre-scan training and the level of stimulation for that session. The remaining nine fMRI runs consisted of the experimental task with the completely crossed cue-stimulation paradigm. Participants were unaware of the different types of runs, and performed the same actions for all the runs. Only data from the nine experimental runs were used for further analyses.

The somatic pain sessions included 46, 47 and 48°C thermal stimulations and the vicarious pain sessions included previously published (*Jackson et al., 2005*) low, medium and high unpleasant images (which were independently normed through a survey with n = 20 participants). The somatic and vicarious pain sessions were conducted on different days to reduce learning effects. During both the conditioning (2 runs) and experimental runs (9 runs), participants received a cue-stimulus pair for each trial and were asked to make a rating on a visual analog scale (same as the calibration session) about the sensation they felt after each trial. For the somatic pain session, the participants rated the intensity of pain they felt during each trial. For the vicarious pain session, the participants were instructed to imagine that the injury occurring in the picture displayed was happening to them and rated how much pain they might feel in that situation.

Each experimental run contained 9 trials (81 total for each session), which were counterbalanced between runs for each participant using a Latin Square design. Experimental trials (i.e., post conditioning) began with a 2s cue followed by a systematic jitter separating the cue from stimulation (i.e., 5, 7, 11s). Participants then received stimulation (somatic or vicarious pain) for 11s followed by a jittered fixation (2, 6 or 14s for somatic pain and 3, 6 or 12s for vicarious pain). The 11s trial duration for somatic pain included a 2s ramp-up, 2s ramp-down and 7s at target temperature; the vicarious pain stimuli remained on the screen for 11s. Finally, participants had 4s to make a rating of the sensation they experienced for the stimulation on a visual analog scale using a trackball (responses were confirmed with a button-click). There was an inter-trial jittered fixation (1, 4, or 10s for somatic pain and 1, 2 or 4s for vicarious pain). The jittered fixations were counterbalanced across trials within a run, so that all runs were of equal duration for each session.

## Behavioral analyses

Participants rated the sensation they felt at every trial during the fMRI sessions. All ratings were averaged across trials and runs for each participant, separately for each level of stimulation—low, medium and high, and for each body site (i.e., upper limb or UL and lower limb or LL). Averaged ratings for all participants from the somatic pain and vicarious pain sessions were tested for differences in stimulation level, and body site separately. All behavioral data were analyzed using the R Project for Statistical Computing (http://www.r-project.org/).

## Image acquisition

Structural images were acquired using high-resolution T1 spoiled gradient recall images (SPGR) for anatomical localization and warped to Montréal Neurological Institute (MNI) space (*Evans et al.,*

*1993*). Functional images were acquired with an echo-planar imaging sequence (TR = 1300 ms, TE = 25 ms, field of view = 220 mm, 64x64 matrix, 3.4 x 3.4 x 3.4 mm$^3$ voxels, 26 interleaved slices with ascending acquisition, parallel imaging with an iPAT acceleration of 2). The somatic pain sessions had 11 runs, which each lasted 5 min and 50s (265 TRs), and the vicarious pain sessions had 11 runs, which each lasted 5 min and 22s (244 TRs). Stimulus presentation and behavioral data acquisition were controlled using Matlab software (MATLAB, The MathWorks, Inc., Natick, Massachusetts, United States).

## Image preprocessing

All images were preprocessed using SPM8 (Wellcome Trust Centre for Neuroimaging, London, UK). Mean structural T1-weighted images were computed for each participant from all imaging sessions. The mean structural images were then co-registered to the first functional image for each participant with an iterative procedure of automated registration using mutual information from the co-registration in SPM8 and manual adjustment of the automated algorithm's starting point until the automated procedure provided satisfactory alignment, and were normalized to MNI space using SPM8, interpolated to $2\times2\times2$ mm$^3$ voxels.

Functional images were corrected for slice-acquisition-timing and motion using SPM8. They were then warped to SPM8's normative atlas using warping parameters estimated from co-registered, high resolution structural images, interpolated to $2\times2\times2$ mm$^3$ voxels, and smoothed with a 8 mm full-width-at-half-maximum (FWHM) Gaussian kernel. Spatial smoothing was performed for between-participant analyses as it does not diminish the sensitivity of multivariate analyses (*Op de Beeck, 2010*), and also captures mesoscopic patterns that are consistent across participants. No spatial smoothing was performed for any of the within-participant analyses.

Prior to preprocessing of functional images, global outlier time points (i.e., 'spikes' in signal) were identified by computing both the mean and the standard deviation (across voxels) of values for each image for all slices. Mahalanobis distances for the matrix of slice-wise mean and standard deviation values (concatenated) were computed for all functional volumes (time), and any values with a significant $\chi^2$ value (corrected for multiple comparisons) were considered outliers (less than 1% of images were outliers). The output of this procedure was later used as a covariate of noninterest in the first level models.

## Imaging analyses

First-level general linear model (GLM) analyses were conducted in SPM8. The first 6 volumes of each run were discarded, and the nine experimental runs were concatenated for each participant (the first two conditioning runs were excluded). Boxcar regressors, convolved with the canonical hemodynamic response function, were constructed to model periods for the 2s cue presentation, the 5, 7, or 11s variable pre-stimulus fixation period, the 11s somatic stimulation (9 levels), and 4s rating periods. The fixation cross epoch was used as an implicit baseline. A high-pass filter of 224s was used for the somatic pain session and 309s was used for the vicarious pain sessions. These values for the high pass filter were determined based on a first-level analysis on the two conditioning runs, where the variance inflation factor was determined to be less than 5%. Contrasts of interest included the low, medium and high stimulation period collapsed across cues and body site (i.e., upper limb and lower limb), upper limb and lower limb contrasts for three levels of stimulation, and overall averaged stimulation collapsed across levels, cues and body site. These contrasts were computed for both the somatic pain and vicarious pain sessions, and were used for the signature response analyses described below.

## Univariate general linear model (GLM) analyses

A second level one-sample *t*-test was calculated for each contrast using robust regression. Specifically, we conducted group-level analyses for somatic pain collapsed across cue, stimulation level and body site against baseline (corrected at FDR q<0.01). We performed an identical group-analysis for vicarious pain against baseline (corrected at FDR q<0.01). We then used these results to identify commonly activated regions (by masking the overlapping significant voxels) for both somatic and vicarious pain.

## Neurologic pain signature (NPS) response analyses

The contrast images from the first-level analyses for each participant were used as the input for the signature response analyses. The weight map from the Neurologic Pain Signature (NPS) pattern (*Wager et al., 2013*) was applied to the contrast images, and a single signature response value was computed for each participant, based on the dot product of the pattern of NPS weights with the contrast image for the participant. These signature response values were then tested for differences between levels of stimulation, body site, and pain modality (i.e., somatic pain or vicarious pain).

## Vicarious pain pattern (VPS) classification

Data from the vicarious pain session for the 30 participants were used to train a LASSO-PCR classifier to predict behavioral responses to different intensities of vicarious pain stimulation, with a leave-one-participant-out cross-validation scheme (*Wager et al., 2013*). The scheme trains the classifier on $N - 1$ participants and generates a weight map that best predicts the behavioral responses, and tests the classification on the left-out ($N^{th}$) participant in order to obtain the cross-validated signature response value for that participant. This process is continued until all participants have been left out of the classification algorithm once to obtain their respective cross-validated signature response values. To display the brain regions that significantly contributed to the prediction, the pattern map was thresholded using a bootstrap procedure with 5000 samples and FDR correction of q<0.05.

## Vicarious pain signature response analyses

The contrast images from the first-level analyses for each participant were used to compute signature responses for the vicarious pain pattern. The weight map from the VPS was applied to the contrast images and a single signature response value was computed for each participant, based on the dot product of the pattern of VPS weights with the contrast image for the participant. These signature response values were then tested for differences between levels of stimulation, body site, and pain modality (i.e., somatic pain or vicarious pain).

## Between-participant local pattern classification

The univariate GLM analyses identified commonly activated regions for both somatic and vicarious pain, namely anterior insula [aINS] and dorsal anterior cingulate cortex [dACC] (*Singer et al., 2004*). These regions were identified by masking the overlapping significant voxels, and were used as regions of interest for separate region-specific local pattern classification. The LASSO-PCR classifier was trained within each of these regions for somatic and vicarious pain. Cross-validated pattern responses within-modality, and cross-predicted pattern responses across modality were then computed for each participant.

## Within-participant local searchlight pattern classification

In order to identify additional regions that might share representations across somatic and vicarious pain, a within-participant local pattern-based searchlight classification analysis was performed on the unsmoothed single trial data using 8 mm spherical searchlights around center voxels (*Kriegeskorte et al., 2006*). A local region classifier (i.e., LASSO-PCR) was trained to predict behavioral ratings from each modality separately for each of the 181,129 local regions, and the pattern obtained was used to cross-validate (within-modality) and cross-predict (between-modality) for each pain modality (i.e., somatic vs. vicarious). For each of the cross-validation folds, one run with upper limb stimulation and one run with lower limb stimulation was excluded from the analysis. The resulting pattern was tested on these left-out runs (cross-validation), and the corresponding trial numbers from the other modality (cross-prediction). The within-modality patterns loosely corroborate the global patterns, but are noisier and have considerably less statistical power.

## Stimulus and anticipation-related activity of NPS and VPS responses

In order to potentially identify activation at time points that do not match the rise of pain experience (including anticipation), a smoothed finite-impulse response model (sFIR) was estimated across all trials for both somatic and vicarious pain (*Lindquist et al., 2009*). These models provide an empirical picture of the time-courses for the different events in a trial (*Lindquist et al., 2009*). The first-level

betas for each TR were estimated separately at the 11s stimulation onset, and at the 2s cue onset, for a total period of 32.5s after onset (which was equivalent to 25 TRs). Then, the NPS and VPS patterns were applied to the image obtained at each time point, and the signature responses were plotted for each of the three levels of stimulation.

### Relationship between the VPS and other types of affective patterns

The relationship between the VPS and patterns identified for (a) romantic rejection (*Woo et al., 2014*) and (b) negative emotion induced by affective pictures (*Chang et al., 2015*), two other types of aversive experience, was tested by computing whole-brain spatial correlations between the patterns of the signatures.

### Upper limb versus lower limb pattern classification

To determine whether a shared somatotopic representation exists for somatic and vicarious pain, a support vector machine (SVM) classifier was trained to differentiate between upper limb and lower limb sites separately for somatic and vicarious pain. The weights obtained following classification of upper limb from lower limb in one modality (e.g., somatic pain) were tested to differentiate between the same body sites in the other modality (e.g., vicarious pain).

## Study 2

### Participants

Twenty-eight right-handed female healthy participants ranging from 35 to 55 years (Mean Age = 44.32 years, SD = 4.58 years; Mean Education Level = 14.75 years, SD = 4.78 years) were recruited for Study 2 (data from 1 participant was discarded, see below). A complete medical interview was carried out to exclude participants with relevant medical or neurological disorders, history of substance abuse, psychiatric illness or chronic pain complaints. All participants gave written informed consent to participate in the study, which was approved by the research and ethics committee of the Autonomous University of Barcelona.

### Mechanical pain stimulation

Mechanical (pressure) stimulation was delivered using a hydraulic device capable of transmitting controlled pressure to a 1 cm$^2$ surface placed on the participants' right thumbnail. Similar to other studies (*Gracely et al., 2004*; *Gracely et al., 2002*; *López-Solà et al., 2010b*), this system consisted of a semi-hard rubber probe attached to a hydraulic piston that was displaced by mechanical pressure. In a calibration session, each participant was trained to report pain intensity and unpleasantness to different pressure stimuli ranging from 2 to 9 kg/cm$^2$ (or up to tolerance threshold) using a numerical rating scale (NRS) ranging from 0 (not at all painful/unpleasant) to 100 (worst pain imaginable/most unpleasant imaginable).

Participants were exposed to two pressure stimulation intensities: (a) low pressure intensity (4.5 kg/cm$^2$ for 10s) and (b) high pressure intensity, which was individually adjusted to provoke severe (above 60 in the NRS), but tolerable, pain for each participant (average 5.9 ± 0.48 kg for 10s). In a preliminary session before the *f*MRI experiment, each participant was familiarized with the two stimulation intensities.

All participants rated perceived pain intensity of the stimuli (low and high) that was later applied during the *f*MRI experiment using a numerical rating scale (NRS) ranging from 0 (no pain) to 100 (the worst pain possible). The 'low' stimulus intensity (4.5 kg/cm$^2$) was capable of producing low-to-moderate pain for all participants during the pre-scan assessment (mean ± SD in the NRS = 36.85 ± 20.38 points), whereas the 'high' stimulus intensity was capable of producing severe but tolerable pain during the pre-scan assessment (mean ± SD in the NRS = 63.50 ± 17.73 points).

### Experimental sessions

All participants were first exposed to the low stimulus intensity task, which was followed by the high stimulus intensity task, occurring approximately 10 min later. A block design *f*MRI paradigm was used consisting of three events per stimulation cycle repeated 5 times: a rest event with pseudorandom variable duration (duration range: 20 to 32s), a 6s anticipatory event starting with a brief auditory stimulus (600ms tone) that cued the subsequent pain condition, and the actual pain event

involving the application of the pressure stimulus (either low or high pressure) for 10s. Each participant was asked to rate pain intensity and unpleasantness immediately after the end of the trial using the NRS described earlier.

### Image acquisition

Participants were scanned on a Philips Achieva 3.0 TX system (Philips Healthcare, Best, The Netherlands), with an eight-channel phased-array head coil and single-shot echoplanar imaging (EPI). Functional sequences consisted of gradient recalled acquisition in the steady state (TR = 2000 ms; TE = 35 ms; flip angle= 90°; dummy volumes = 4) within a field of view of 23 cm, a 96×69 matrix, and slice thickness of 4mm (inter-slice gap, 1 mm). Twenty-two slices parallel to the anterior-posterior commissure covered the whole-brain.

### Image preprocessing

Imaging data were processed using MATLAB version 2011b (The MathWorks Inc, Natick, Mass) and Statistical Parametric Mapping software (SPM8; Wellcome Trust Centre for Neuroimaging, London, UK). Preprocessing involved motion correction, spatial normalization and smoothing using a Gaussian filter (full-width at half-maximum, 8 mm). Data were normalized to the standard SPM-EPI template and resliced to 2 mm isotropic resolution in MNI space. From the original sample of 29 participants, data from 1 participant was excluded due to poor signal in the frontal lobe and excessive head movement (z-axis translation > 2.0 mm). Translation and rotation estimates (*x*, *y*, *z*) were less than 2 mm or 2°, respectively, for all the included participants.

### Imaging analyses

The GLM as implemented in SPM8 was used to estimate brain responses to pain for each participant. Separate regressors for the anticipatory and the pain periods were used, considering a hemodynamic delay of 4s. In two previous studies using similar procedures (*López-Solà et al., 2010b*; *Pujol et al., 2009*), the duration of brain responses to 10s pressure stimuli of similar intensity was systematially oberved to extend to 16 s (average response duration across pain processing regions), which was consistent with previous studies (*Cauda et al., 2014*; *Wager et al., 2013*). Pain-related activation was modeled using a pain condition of 16s duration. Autocorrelation was not modeled to avoid dramatic decreases in efficiency when autocorrelation is misestimated. Pressure stimulation – baseline contrast images were calculated for each participant.

### Signature response analysis

The contrast images from the first-level analyses for each participant were used as the input for the signature response analyses. The weight map from the Neurologic Pain Signature (NPS) pattern (*Wager et al., 2013*) was applied to the contrast images and a single signature response value was computed for each contrast image for each participant, based on the dot product of the pattern of NPS weights with the contrast image for the participant. These signature response values were then tested for differences between levels of stimulation.

## Study 3

### Participants

Eighteen healthy participants were recruited through the Icahn School of Medicine at Mount Sinai (ISMMS). Participants were excluded based on medical illness or history in first-degree relatives of developmental disorders, learning disabilities, autism, affective disorders, and anxiety disorders. Participants with a history of substance or alcohol dependency or abuse within one year prior to participation were excluded as well. There were a total of 14 right-handed and 1 left-handed male participants (measured by the Edinburgh Inventory Handedness Questionnaire (*Oldfield, 1971*). All participants provided written informed consent, approved by the ISMMS Institutional Review Board.

## Experimental sessions

### Anticipatory stimulation paradigm (electrical or shock pain)

Electrodermal stimulations (shock) were placed to the left extremity, with two MRI compatible EL508 disposable electrodes pre-gelled with a 1-cm diameter circular contact area. This paradigm used uncomfortable electrodermal stimulation (low current) (LaBar et al., 1998) applied to the left extremity (lower medial side above the ankle). This is an S1-S2 paradigm with 1s of S1 for safe/stimulation cue and 1s of S2 with/without stimulation, with an inter stimulus interval (ISI) of 3s. The inter trial interval (ITI) was jittered (4–12s, exponentially distributed, mean = 5s). Mean duration for each trial was 10s. Different colored squares, identical in shape, size, and luminance, served as the cue and target stimuli. Participants were instructed to view visual stimulus squares (red, blue, yellow, or green) presented pseudo-randomly. Participants were informed that a red square with the word 'STIMULATION', when presented, indicates that stimulation may follow when a yellow square appears. They were also informed that a blue square with the word 'SAFE' indicates that there was no stimulation when a yellow square appears. Each run had 48 counterbalanced trials with 24 safe trials and 24 stimulation trials. Electrodermal stimulation was given during 50% of the S2 period of the stimulation trials (12 of 24 trials) when the 'STIMULATION' stimulus preceded the yellow square. In addition, there were 4 catch trials with the word 'PRESS' to instruct the participant to press a button to ensure that participants were watching the screen. There was a 30s resting period at the beginning and end of each run. Each run took 580s or about 10 min. Two runs of the shock pain paradigm were conducted. Due to the fact that the threshold for pain perception varies markedly among individuals, the intensity was decided and recorded individually before scanning at a medium level of pain tolerable by the participant (using a 5-point scale). No active task was required for shock or safe trials. However, participants were instructed to press a button when they saw the target following the 'PRESS' cue in order to maintain their attention during scanning. Pain intensity and unpleasantness ratings by each participant were also collected at the end of each run with a 5-point scale using a glove response device.

### Empathy for others' pain (observed pain)

Participants were presented with 256 photographs (in color) of hands or feet of individuals in painful or non-painful situations, and were asked to judge whether the person shown in the image was suffering from pain or not (Gu et al., 2010; Gu et al., 2013). Stimuli were jittered and presented in an event-related fMRI design such that presentation of each type of picture was counterbalanced and pseudo-randomized, with each trial having all events proceeded and followed equally. Each 5.5s trial consisted of a presentation of a stimulus picture, along with the two response categories (i.e., no pain and pain) for 2.5s, during which participants made their responses. This was followed by 3s of fixation. Sixty-four images were selected for presentation in each run, for a total of four runs. The order of trials in each block/run was randomized with optimized efficiency. There was a 30s fixation period at the beginning and end of each run to allow the skin conductance and hemodynamic responses to return to baseline.

## Image acquisition

All MRI acquisitions were obtained on a 3 Tesla Siemens Allegra MRI system at the Icahn School of Medicine at Mount Sinai. All participants underwent only one session with all scanning sequences. The whole scan session lasted about one and a half hours. Foam padding was used to keep participants' heads still. All images were acquired along axial planes parallel to the anterior commissure-posterior commissure (AC-PC) line. The order and scan sequences were as follows: (1) A low resolution sagittal high speed scout image; (2) A high-resolution T2-weighted anatomical volume of the whole brain, acquired on an axial plane parallel to the AC-PC line with a turbo spin-echo (TSE) pulse sequence with the following parameters: 40 axial slices of 4 mm thick, skip=0 mm, TR = 4050 ms, TE = 99 ms, flip angle=170°, field of view (FOV)=240 mm, matrix size=448×512, voxel size=$0.47×0.47×4$ mm$^3$; (3) T2*-weighted images. Slices were obtained corresponding to the T2-weighted images. The fMRI imaging was performed using a gradient echoplanar imaging (GE-EPI) sequence: 40 axial slices, 4 mm thick, and skip=0 mm, TR = 2500 ms, TE = 27 ms, Flip angle = 82°, FOV=240 mm, and matrix size=64×64. Each run started with 2 dummy volumes before the onset of the task to allow for equilibration of T1 saturation effects. Stimulation events were modeled as single

trials. The order of trials in each block/run was randomized with optimized efficiency. A total of two EPI runs with 232 image volumes per run for the shock paradigm and a total of four EPI runs with 165 image volumes per run for the observed pain paradigm were acquired for each participant.

### Image preprocessing

The functional scans were adjusted for slice timing, realigned to the first volume, coregistered to the T2 image, normalized to a standard MNI template, and spatially smoothed with an 8 mm FWHM Gaussian kernel.

### Imaging analyses

Event-related analyses of the fMRI data from the two tasks were conducted using SPM8 (Wellcome Trust Centre for Neuroimaging, London, UK). GLM (*Friston et al., 1995*) for the functional scans from each participant was performed by modeling the observed event-related activation (*Todd et al., 2013*) and regressors that identified the relationship between task events and the hemodynamic response. Regressors were created by convolving a train of delta functions representing the sequence of individual events with the default SPM basis function, which consists of a synthetic hemodynamic response function composed of two gamma functions (*Friston et al., 1998*). For shock pain, there were six regressors (all cues, stimulation cues, all targets, targets with stimulation cues, stimulation, and catch trials); for observed pain, there were two regressors (painful, non-painful). Six parameters generated during motion correction were entered as covariates. Shock pain – baseline, and observed pain – baseline contrast images were calculated for each participant.

### Signature response analysis

The contrast images from the first-level analyses for each participant for both shock and observed pain conditions were used to compute signature responses. The weight map from the Neurologic Pain Signature (NPS) pattern (*Wager et al., 2013*), and the Vicarious Pain Signature (VPS) were applied separately to the contrast images and signature response values were computed for each contrast image for each participant. These signature response values were then tested for differences between types of stimulation, for both the NPS and VPS.

## Acknowledgements

We would like to thank the members of the Cognitive and Affective Neuroscience Lab for helpful discussions, and personnel at the Intermountain Neuroimaging Consortium for help with fMRI data collection for Study 1. Matlab code implementing analyses described in this work is freely available from http://wagerlab.colorado.edu.

## Additional information

### Funding

| Funder | Grant reference number | Author |
|---|---|---|
| National Institutes of Health | R01MH076136 | Tor D Wager |
| National Institutes of Health | R01DA035484 | Tor D Wager |
| National Institutes of Health | R21MH083164 | Jin Fan |

The funders had no role in study design, data collection and interpretation, or the decision to submit the work for publication.

### Author contributions

AK, Conception and design, Acquisition of data, Analysis and interpretation of data, Drafting or revising the article; C-WW, Analysis and interpretation of data, Drafting or revising the article; LJC, Interpretation of data, Drafting or revising the article; LR, Conception and design, Analysis and interpretation of data; XG, For Study 3, Conception and design, Acquisition of data, Drafting or revising the article; ML-S, For Study 2, Conception and design, Acquisition of data, Drafting or revising the

article; PLJ, Provided stimuli for vicarious pain stimuli, Drafting or revising the article, Contributed unpublished essential data or reagents; JP, For Study 2, Conception and design, Drafting or revising the article; JF, For Study 3, Conception and design, Drafting or revising the article; TDW, Conception and design, Analysis and interpretation of data, Drafting or revising the article

### Author ORCIDs

Anjali Krishnan, http://orcid.org/0000-0002-4741-9103

Jin Fan, http://orcid.org/0000-0001-9630-8330

### Ethics

Human subjects: All participants for Study 1 provided informed consent, and all experimental procedures were conducted with the approval of the Institutional Review Board of the University of Colorado Boulder. All participants for Study 2 provided informed consent, and all experimental procedures were conducted with the approval of the research and ethics committee of the Autonomous University of Barcelona. All participants for Study 3 provided informed consent, and all experimental procedures were conducted with the approval of the Icahn School of Medicine at Mount Sinai Institutional Review Board.

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

## Appendix

We undertook several additional analyses to test the dissociability of somatic and vicarious pain patterns. Here we report results for analyses that have been referenced in the main text, and include an extended discussion in support of our findings.

### Positive and negative pain-relevant signature response analyses

We ran additional analyses to rule out the possibility that the NPS responses for vicarious pain were being driven by specific regions within the NPS mask (e.g., the visual regions). Using a mask obtained from Neurosynth.org (*Yarkoni et al., 2011*), we performed feature selection for the NPS pattern with 'pain' as the search term. We obtained a reverse inference map based on a meta-analysis for 'pain' from the journal articles coded in the Neurosynth database ($N$ = 224). The reverse inference map indicates regions that are most likely associated with journal articles that focus on 'pain' than any other topic. The reason that visual regions are included in the NPS mask is because based on the Neurosynth definition of reverse inference, these regions are least likely to be associated with journal articles that focus on pain, and therefore have negative weights in the NPS mask. In the case where the NPS mask is applied to data that include visual stimuli (e.g., vicarious pain), activation of the visual regions of the brain are correlated with the visual regions of the NPS mask. Therefore, we extracted the NPS weights associated with either only the positive or only the negative regions of the Neurosynth 'pain' mask and computed the signature response for somatic and vicarious pain for data from Study 1.

The removal of the negative regions of the Neurosynth 'pain' mask had no effect on the signature response for somatic pain (see *Appendix 1—figure 1A*; $t_{UL}(27)$ = 9.26, p<0.0001, $t_{LL}(27)$ = 7.18, p<0.0001 for somatic pain; $t_{UL}(27)$ = −0.56, *n.s.*, $t_{LL}(27)$ = 0.046, *n.s.*, for vicarious pain), but greatly reduced the signature response for vicarious pain. In contrast, the weights corresponding to the negative regions of the Neurosynth "pain" mask showed increased activity for vicarious pain (although still did not detect the difference in intensity), but greatly reduced the signature response for somatic pain (see *Appendix 1—figure 1B*; $t_{UL}(27)$ = −5.22, p<0.005, $t_{LL}(27)$ = −2.81, p<0.01 for somatic pain; $t_{UL}(27)$ = −1.14, *n.s.*, $t_{LL}(27)$ = −3.05, p<0.01 for vicarious pain). Limiting the NPS to only regions that are positively related to pain in previous studies [based on (*Yarkoni et al., 2011*)], resulted in strong NPS responses to somatic pain but no responses to any vicarious pain conditions. Critically, however, the NPS did *not* predict vicarious pain intensity, suggesting that the two types of pain may engage fundamentally different circuits. This suggests that the negative NPS response for vicarious pain was mostly driven by weights in the negative regions of the Neurosynth 'pain' mask.

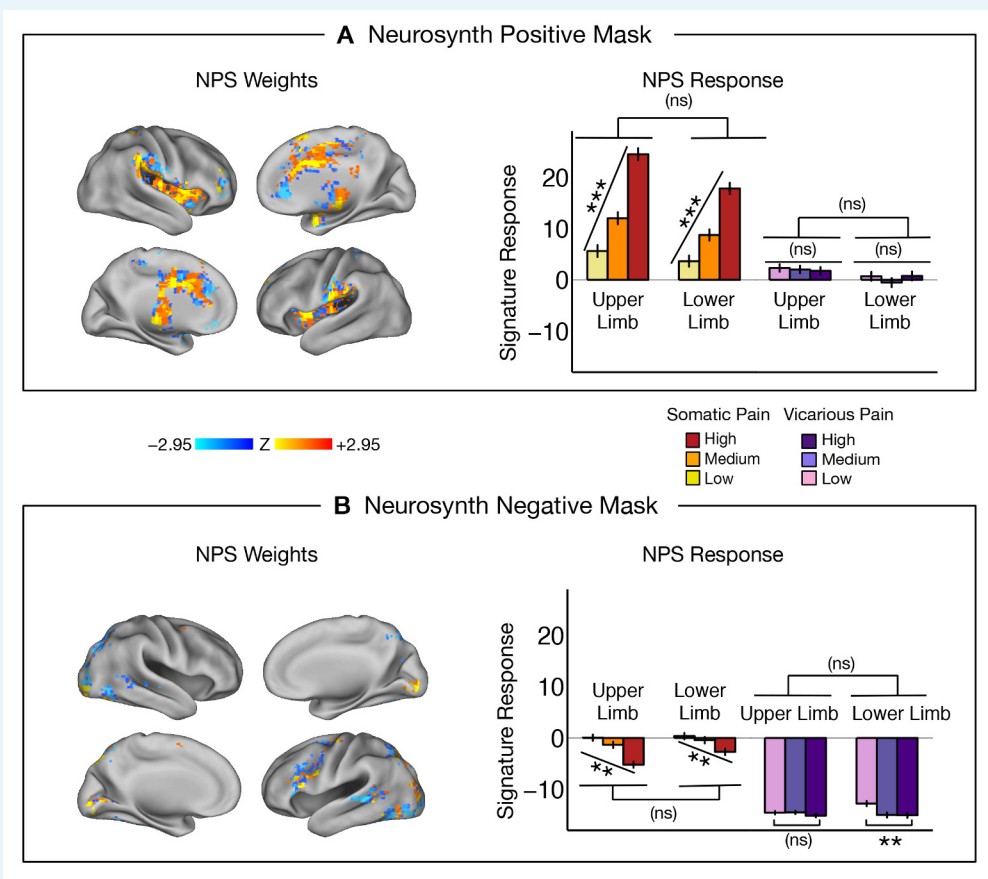

**Appendix 1—figure 1.** NPS responses in (**A**) (Left): NPS weights from positive regions of the Neurosynth mask for 'pain'; **A** (Right): Signature response computed as the dot product of the NPS pattern weights and estimated activation maps for each participant (including within-participant standard error of the mean; $t_{UL}(27) = 9.26$, p<0.0001, $t_{LL}(27) = 7.18$, p<0.0001 for somatic pain; $t_{UL}(27) = -0.56$, n.s., $t_{LL}(27) = 0.046$, n.s., for vicarious pain); **B** (Left): NPS weights from negative regions of the Neurosynth mask for 'pain'; **B** (Right): Signature response computed as the dot product of the NPS pattern weights and estimated activation maps for each participant (including within-participant standard error of the mean; $t_{UL}(27) = -5.22$, p<0.005, $t_{LL}(27) = -2.81$, p<0.01 for somatic pain; $t_{UL}(27) = -1.14$, n.s., $t_{LL}(27) = -3.05$, p<0.01 for vicarious pain).

## Re-training of a new somatic pain predictive pattern

In addition to testing the NPS—which was defined *a priori* based on previous studies—we used LASSO-PCR (Least Absolute Shrinkage and Selection Operator-regularized Principal Components Regression) to identify a somatic pain-predictive pattern using cross-validated analyses on data from Study 1. This ensured that the data and training procedures used to identify somatic and vicarious pain patterns were identical. Data from the somatic pain session for 28 participants were used to train the LASSO-PCR classifier to predict behavioral responses to different intensities of somatic pain stimulation, with leave-one-participant-out cross-validation (*Wager et al., 2013*). The pattern map was thresholded using a bootstrap procedure with 5000 samples and FDR correction of q<0.05 (see *Appendix 1—figure 2A*).

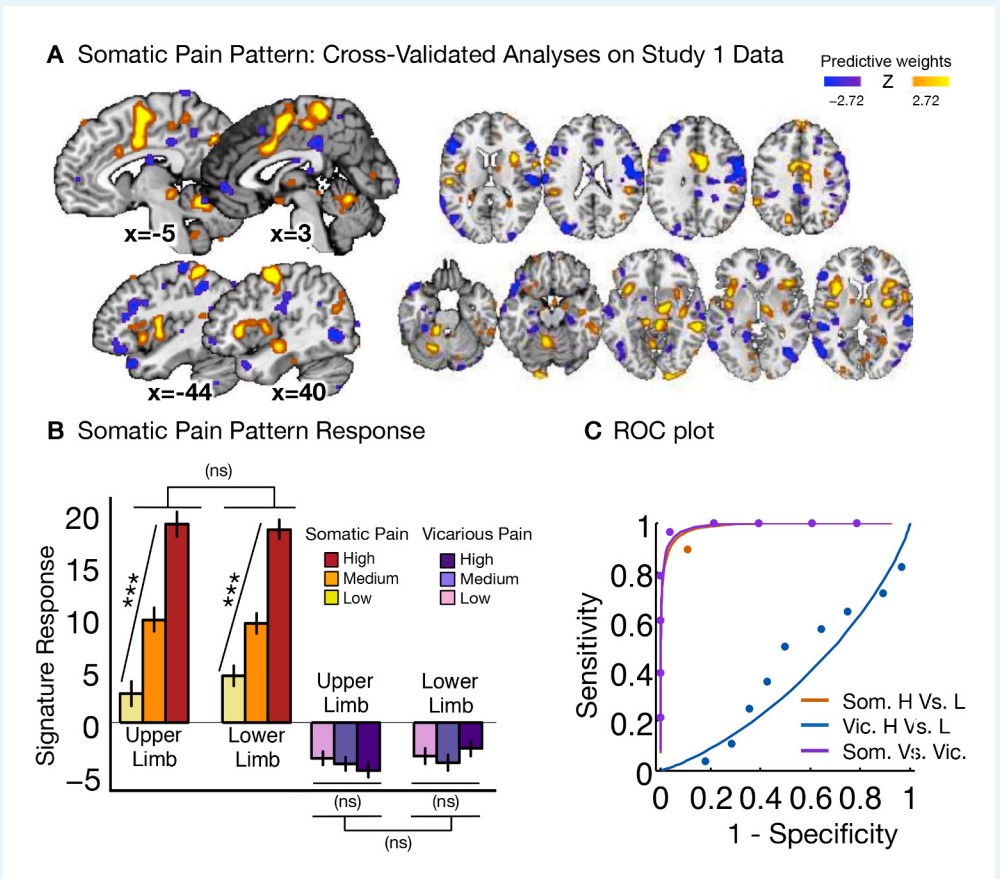

**Appendix 1—figure 2.** Somatic pain pattern and analyses. (**A**) Thresholded LASSO-PCR (Least Absolute Shrinkage and Selection Operator-regularized Principal Components Regression) derived pattern for somatic pain; **B** Signature response computed as the dot product of the somatic pain pattern weights and estimated activation maps for each participant (including within-participant standard error of the mean; $t_{UL}(27) = 8.06$, p<0.0001, $t_{LL}(27) = 9.20$, p<0.0001, $t_{UL-LL}(27) = -0.17$, n.s., for somatic pain; $t_{UL}(27) = -1.35$, n.s., $t_{LL}(27) = 0.66$, n.s.; $t_{UL-LL}(27) = -1.04$, n.s., for vicarious pain); **C** Receiver Operator Characteristic (ROC) plot for two-choice forced-alternative accuracies for somatic and vicarious pain, high and low somatic pain, and high and low vicarious pain (accuracy$_{Som-Vic}$ = 89%, p<0.0001, accuracy$_{Hsom-Lsom}$ = 96%, p<0.0001; accuracy$_{Hvic-Lvic}$ = 50% n.s.).

The cross-validated signature response monotonically increased for each level of somatic pain for both the upper limb and lower limb sites, with no difference between body sites ($t_{UL}(27)$ = 8.06, p<0.0001, $t_{LL}(27)$ = 9.20, p<0.0001, $t_{UL-LL}(27) = -0.17$, n.s.; see **Appendix 1—figure 2B**). The somatic pain pattern significantly differentiated between somatic and vicarious pain, and high versus low somatic pain in a two-choice forced-alternative test (accuracy$_{Som-Vic}$ = 89%, p<0.0001, accuracy$_{Hsom-Lsom}$ = 96%, p<0.0001; see **Appendix 1—figure 2C**), but did not differentiate between high and low levels of vicarious pain ($t_{UL}(27) = -1.35$, n.s., $t_{LL}(27)$ = 0.66, n.s.; $t_{UL-LL}(27) = -1.04$, n.s.; accuracy$_{Hvic-Lvic}$ = 50% n.s.). Thus, the somatic pain pattern identified in this study is comparable to the NPS, an established signature pattern than has been shown to track somatic pain successfully across studies.

## Lack of VPS dependence on visual cortical processing

An important issue is the degree to which the differences between the VPS and NPS are driven by differences in the sensory modality of stimulation per se. This concern applies to whole-brain analyses but not to local analyses of dACC and aINS, which are heteromodal, multisensory regions thought to encode 'pain affect' and other modality-independent motivational processes. The predictive voxels in the VPS map are clustered in sensory modality-independent 'mentalizing' systems, rather than visual processing systems. However, to further test the (in)dependence of vicarious pain prediction on visual processing, we re-trained the VPS excluding the entire occipital cortex. We used a LASSO-PCR classification to predict behavioral responses to different intensities of vicarious pain stimulation, with a leave-one-participant-out cross-validation scheme, identical to the original analysis performed with whole brain data. The pattern map with occipital cortex removed was thresholded using a bootstrap procedure with 5000 samples and FDR correction of q<0.05 (see *Appendix 1—figure 3A*).

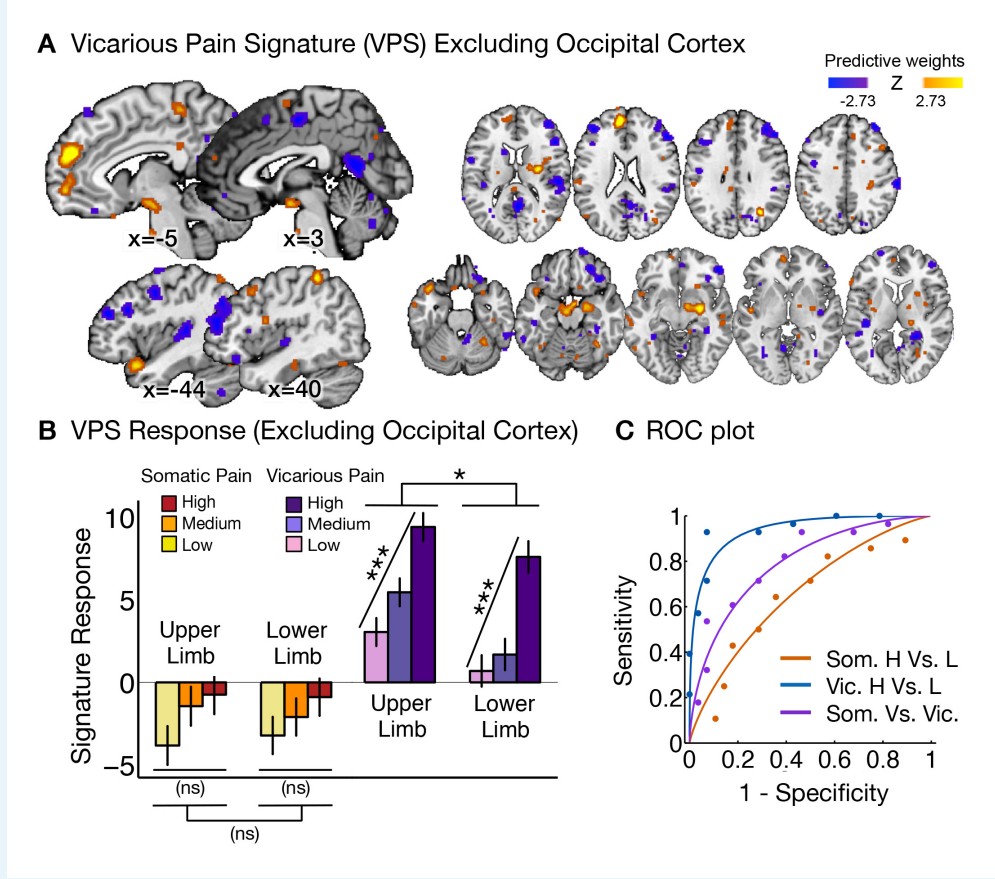

**Appendix 1—figure 3.** Vicarious pain pattern (excluding occipital cortex) and analyses. (**A**) Thresholded LASSO-PCR (Least Absolute Shrinkage and Selection Operator-regularized Principal Components Regression) derived pattern for vicarious pain excluding the occipital cortex; (**B**) Signature response computed as the dot product of the VPS pattern weights and estimated activation maps for each participant (including within-participant standard error of the mean; $t_{UL}(27) = 5.75$, p<0.0001, $t_{LL}(27) = 5.45$, p<0.0001; $t_{UL-LL}(27) = 2.21$, p<0.05, for vicarious pain; $t_{UL}(27) = 1.56$, n.s., $t_{LL}(27) = 1.23$, n.s.; $t_{UL-LL}(27) = 0.03$, n.s., for somatic pain); (**C**) Receiver Operator Characteristic (ROC) plot for two-choice forced-alternative accuracies for vicarious and somatic pain, high and low vicarious pain, and high and low somatic pain (accuracy$_{Vic-Som}$ = 71%, p<0.05; accuracy$_{Hvic-Lvic}$ = 93%, p<0.0001; accuracy$_{Hsom-Lsom}$ = 64%, n. s.).

The signature response (excluding the occipital cortex) monotonically increased for each level of vicarious pain for both the upper limb and lower limb sites ($t_{UL}$(27) = 5.75, p<0.0001, $t_{LL}$(27) = 5.45, p<0.0001; see *Appendix 1—figure 3B*). In addition, there was a marginal significant difference between upper limb and lower limb sites ($t_{UL-LL}$(27) = 2.21, p<0.05). The vicarious pain pattern excluding the occipital cortex significantly differentiated between somatic and vicarious pain, and high versus low vicarious pain in a two-choice forced-alternative test (accuracy$_{Vic-Som}$ = 71%, p<0.05, accuracy$_{Hvic-Lvic}$ = 93%, p<0.0001; see *Appendix 1—figure 3C*), but did not differentiate between high and low levels of somatic pain ($t_{UL}$(27) = 1.56, *n.s.*, $t_{LL}$(27) = 1.23, *n.s.*; $t_{UL-LL}$(27) = 0.03, *n.s.*, accuracy$_{Hsom-Lsom}$ = 64%, *n.s.*). The predictive accuracy for vicarious pain was qualitatively identical, further indicating that the brain patterns that predict vicarious pain are not tied to sensory-specific cortical areas.

Based on previous literature, somatic pain was administered on the left limbs, and the visual stimuli used to evoke vicarious pain showed injuries to the right limbs. This mirrors the side of painful stimulation from the observer's point of view in an allocentric reference frame, but not an egocentric one; thus, lateralization of stimulus presentation could potentially determine some of the functional properties of the VPS. To test this, we repeated the signature response analysis with a left-right flipped version of the VPS pattern (excluding the occipital cortex), which preserves the pattern but with opposite laterality. The results remained the same: The flipped VPS pattern did not track somatic pain intensity ($t_{UL}$(27) = 1.47, *n.s.*, $t_{LL}$(27) = 0.04, *n.s.*, accuracy$_{Hsom-Lsom}$ = 50%, *n.s.*), but did track vicarious pain intensity ($t_{UL}$(27) = 2.68, p<0.05, $t_{LL}$(27) = 2.53, p<0.05, accuracy$_{Hvic-Lvic}$ = 79%, p<0.005). These results indicate that the laterality of the VPS, and by extension the laterality of the stimuli it was trained on, are not an important determinant of its functional properties.

## Habituation or sensitization across trials

Another issue that may make the NPS and VPS responses appear more dissimilar is differential habituation or sensitization across trials. To test this, we estimated NPS and VPS amplitudes for each trial using a 'beta series' approach (*Mumford et al., 2012*) and examined their stability across trials. For this analysis, we used single-trial beta estimates for the stimulation period (one per trial), and applied the NPS and VPS patterns. For each participant, we then regressed the NPS responses against the reported somatic pain rating, and obtained the residual activity for somatic pain. We followed the same procedure for each participant for vicarious pain, and regressed the VPS responses against the reported vicarious pain rating to obtain the corresponding residual activity for vicarious pain. We then averaged the responses trial-by-trial across participants, and plotted the dynamic changes for the behavior, predicted signature response, and residual activity for somatic and vicarious pain. We did not see any evidence for systematic variation [e.g., sensitization or habituation; (*Jepma et al., 2013*)] across trials (*Appendix 1—figure 4*).

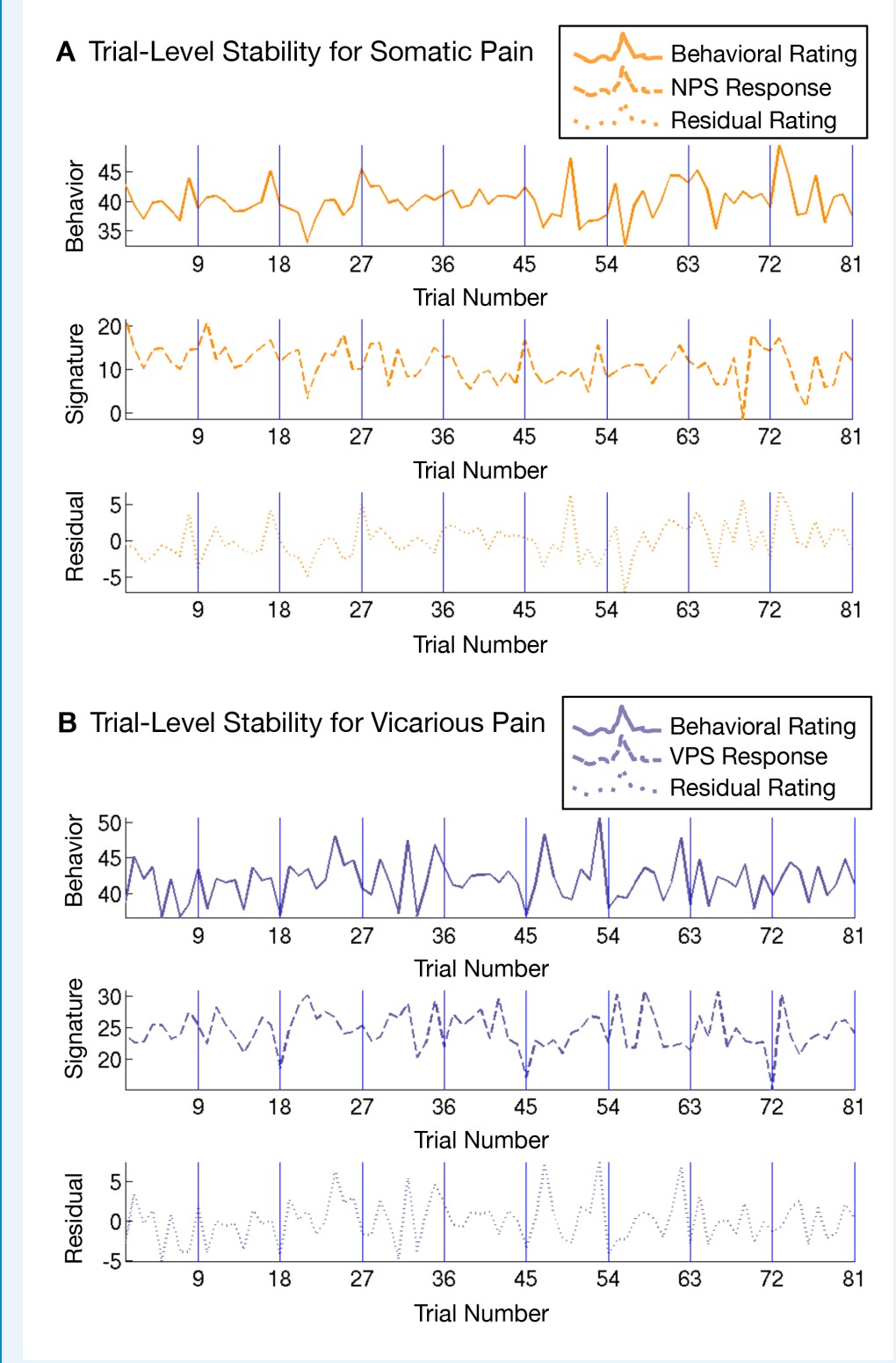

**Appendix 1—figure 4.** Trial-level stability of the NPS and VPS responses across all trials. The first row depicts subjective ratings for somatic pain (top panel) and vicarious pain (bottom panel) respectively, the second row depicts the ratings predicted by NPS (top panel) and VPS (bottom panel) respectively, and the third row shows the residual rating variance not predicted by the NPS (top panel) and VPS (bottom panel) respectively.

## Within participant whole-brain and local classification

Patterns tailored to predict somatic and vicarious pain for each individual might be more accurate, and thus provide better markers for representation that could yield different results. Therefore, we performed within-participant classification of vicarious pain patterns across the whole brain and also within the bilateral aINS and dACC. We used unsmoothed (but corrected for slice-acquisition-time and motion, and warped to SPM's normative atlas) images for these analyses. Single-trial level analyses were conducted where first-level betas for each trial were estimated by modeling the 2s cue presentation, the 5, 7, or 11s variable pre-stimulus fixation period, the 11s stimulation (9 levels) and 4s rating periods, and the fixation cross epoch was used as an implicit baseline. Each of the 9 levels of the 11s stimulation period were modeled trial-by-trial with a total of 81 trial-level betas for each participant per experimental session. Trials with high variance inflation factors (VIFs > 2.5), and unreliable ratings were excluded from further analysis. The estimated betas were used for training classifiers (i.e., LASSO-PCR and SVM) for each participant. For each of the cross-validated iterations, one run with upper limb stimulation and one run with lower limb stimulation were excluded from the analysis. The classifier was trained on the trials from the remaining runs and the resultant pattern was tested both on the left out runs from the training modality (i.e., within modality cross-validation) and all trials from the other modality (i.e., between modality cross-prediction).

We performed a whole-brain LASSO-PCR prediction of each modality's behavioral ratings, cross-validated signature responses within modality, and cross-predicted signature responses between modalities *Appendix 1—figure 5*). The somatic pain maps were able to accurately predict increases in somatic pain ($t(27) = 10.65$, $p < 0.0001$, average accuracy$_{High-Low} = 89\%$, $p < 0.0001$), and the vicarious pain maps were able to strongly predict vicarious pain ($t(27) = 3.37$, $p < 0.005$, average accuracy$_{High-Low} = 71\%$, $p < 0.05$), but importantly, somatic pain patterns could not track vicarious pain ($t(27) = -0.13$, *n.s.*, average accuracy$_{High-Low} = 57\%$, *n.s.*), and vicarious pain patterns could not track somatic pain ($t(27) = 0.93$, *n.s.*, average accuracy$_{High-Low} = 54\%$, *n.s.*). A one-sample *t*-test on individual voxel weights revealed very similar distributed patterns to that observed in the between-participant VPS, which suggests that the patterns are remarkably stable across participants. We note that accuracy values here reflect prediction of single trial responses, and so are not directly comparable with between-participant accuracy values. In addition, the whole brain spatial patterns predictive of somatic and vicarious pain were spatially uncorrelated (mean r = 0.0055 ± 0.0065 S.E., *n.s.*)

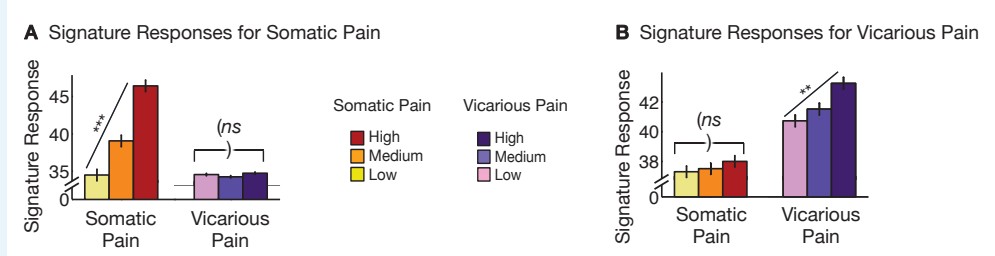

**Appendix 1—figure 5.** Within-participant cross-validated average signature responses. (**A**) Somatic pain pattern response collapsed across body site (including within-participant standard error of the mean; $t_{Hsom-Lsom}(27) = 10.65$, $p < 0.0001$, average accuracy$_{High-Low} = 89\%$, $p < 0.0001$; $t_{Hvic-Lvic}(27) = -0.13$, *n.s.*); (**B**) Vicarious pain pattern response collapsed across body site (including within-participant standard error of the mean; $t_{Hvic-Lvic}(27) = 3.37$, $p < 0.005$, average accuracy$_{High-Low} = 71\%$, $p < 0.05$; $t_{Hsom-Lsom}(27) = 0.93$, *n.s.*).

We also performed ROI-based classification within overlapping regions identified from the GLM analyses, specifically, the bilateral aINS and dACC (smoothed with a half-mm kernel).

Both ROIs were able to differentiate between high and low somatic pain (within modality; aINS accuracy$_{High\text{-}Low}$ = 89%, p<0.0001, dACC accuracy$_{High\text{-}Low}$ = 75%, p<0.01). Vicarious pain however showed marginal classification for high versus low vicarious pain for the aINS (within-modality; average accuracy$_{High\text{-}Low}$ = 71%, p<0.05) and near chance classification for the dACC (average accuracy$_{High\text{-}Low}$ = 57%, n.s.). More importantly, neither pattern within the aINS or the dACC demonstrated cross-modality classification of high versus low stimulation (somatic → vicarious average accuracy: aINS = 46%, n.s., dACC = 54%, n.s.; vicarious → somatic average accuracy: aINS = 50%, n.s., dACC = 36%, n.s.).

Finally, we trained within-participant support vector machines (SVMs) to differentiate between upper and lower limb sites separately for somatic and vicarious pain. The leave-one-participant-out SVM classifier successfully discriminated somatic pain on upper versus lower limb sites (within-participant average accuracy$_{UL\text{-}LL}$= 100%, p<0.0001). SVM classification of vicarious pain also accurately discriminated upper versus lower limb sites (within-participant average accuracy$_{UL\text{-}LL}$= 100%, p<0.0001). Together, these results corroborate the between-participant findings.

## Between-participant local pattern classification

We used commonly activated regions (by masking the overlapping significant voxels) for both somatic and vicarious pain identified by the univariate analyses, namely anterior insula (aINS) and dorsal anterior cingulate cortex (dACC), as regions of interest for local pattern classification (collapsed across body site). We trained the LASSO-PCR classifier within each of these regions for somatic and vicarious pain, cross-validated signature responses within modality, and cross-predicted signature responses between modalities.

We found that the local patterns for somatic and vicarious pain were spatially uncorrelated (aINS: r = −0.02; dACC: r = 0.02). The somatic pain pattern in the bilateral aINS showed a linear increase for somatic pain ($t(27)$ = 7.05, p<0.0001; accuracy$_{High\text{-}Low}$ = 89%, p<0.0001), but did not respond to vicarious pain ($t(27)$ = −0.51, n.s.; accuracy$_{High\text{-}Low}$ = 39%, n.s.). On the other hand, the vicarious pain pattern in the bilateral aINS showed neither within-modality classification ($t(27)$ = −1.14, n.s.; accuracy$_{High\text{-}Low}$ = 68%, n.s.), nor cross-modality classification ($t(27)$ = 1.56, n.s.; accuracy$_{High\text{-}Low}$ = 54%, n.s.). The results from the dACC showed within-modality classification for somatic pain ($t(27)$ = 4.02, p<0.001; accuracy$_{High\text{-}Low}$ = 75%, p<0.01), but no cross-modality classification ($t(27)$ = −0.07, n.s. accuracy$_{High\text{-}Low}$ = 50%, n.s.). The vicarious pain pattern in the dACC showed neither within-modality classification ($t(27)$ = −1.20, n.s.; accuracy$_{High\text{-}Low}$ = 36%, n.s.), nor cross-modality classification ($t(27)$ = −0.14, n.s.; accuracy$_{High\text{-}Low}$ = 50%, n.s.). Together, these results show that shared representations for somatic and vicarious pain representations in the aINS and dACC cannot be captured by local analyses.

## Power and replicability

Power and replicability are issues in every study, particularly when making inferences on selective activation patterns. For example, if the NPS responds to vicarious pain, but with weaker intensity [e.g., as in Hutchinson et al., (*Hutchinson et al., 1999*)], is our study powered to detect differences? We address this issue in three ways.

First, we used a study design that does not require that the levels of activity for somatic and vicarious pain to be the same in order to detect similarity. Thus, if vicarious pain activated the same patterns as somatic pain, but more weakly, we would identify the representations as overlapping as long as there is some measurable increase in the NPS with high vicarious pain. However, our findings do not indicate such sharing.

Second, we use between-participant multi-system predictive , which do not rely on multiple comparisons correction—which dramatically reduces power and replicability (*Button et al., 2013*; *Yarkoni, 2009*)—for valid tests of the population codes. These analyses also allow for unbiased assessments of power to detect cross-modality predictive effects based on the observed strength of the within-modality predictive effects. We performed a formal power analysis for out-of-sample individuals, which indicated that in the NPS, we have 100% power (N needed for 80% power = 4) to detect vicarious pain responses that are as strong as somatic pain responses, and 79.2% power (N needed for 80% power = 29) to detect vicarious pain responses that are only 25% as strong as somatic pain responses. In the VPS, we have 100% power (N needed for 80% power = 4) to detect somatic pain responses that are as strong as vicarious pain responses, and 87.3% power (N needed for 80% power = 24) to detect somatic pain responses that are only 25% as strong as vicarious pain responses.

Third, we demonstrate direct replication in new samples with Study 3, which includes data from a separate sample, scanner, and institution, and show the dissociation of the NPS and VPS responses to somatic and vicarious pain, respectively.

## Upper versus lower limb pattern classification within the somatosensory cortex

We ran additional cross-classification analyses to predict stimulation site (upper versus lower limb), restricting training to only the primary somatosensory cortex. We used a support vector machine (SVM) classifier with data that were masked for SI & SII regions [obtained from the SPM Anatomy Toolbox (*Eickhoff et al., 2005*)] and previously cited regions implicated for somatotopy for somatic pain (*Baumgärtner et al., 2010*). The results revealed strong somatotopy for somatic pain (upper versus lower limb: 96% accuracy, p<0. 0001) and weak somatotopy for vicarious pain (79% accuracy, p<0. 01). Critically, however, cross-classification was at chance for both patterns, and the patterns themselves were spatially uncorrelated (r = 0.022), indicating that vicarious pain does not share somatotopic representations with somatic pain in the somatosensory cortex.

## Laterality effects in upper versus lower limb pattern classification

In the present study, somatic pain was administered to the volar surface of the left forearm and the dorsal surface of the left foot, whereas the visual stimuli for the vicarious pain session showed pictures of injury about the occur to the right hand and right foot. Importantly, the weights in contralateral hemispheres for somatic pain (right) and vicarious pain (left) were spatially uncorrelated (r = 0.07). In order to further account for the difference in laterality between the two modalities, we flipped the contrast images along the *x*-axis for the testing set from left to right and found that the weights obtained from classifying the body sites in one modality still performed below chance on the other modality (see *Appendix 1—figure 6*). Together, these results reveal that somatosensory regions do not show somatotopy for vicarious pain, irrespective of the laterality of stimulation site (right for vicarious pain and left for somatic pain). However, it will be important for future research to examine the somatotopy question with greater precision using more carefully matched designs.

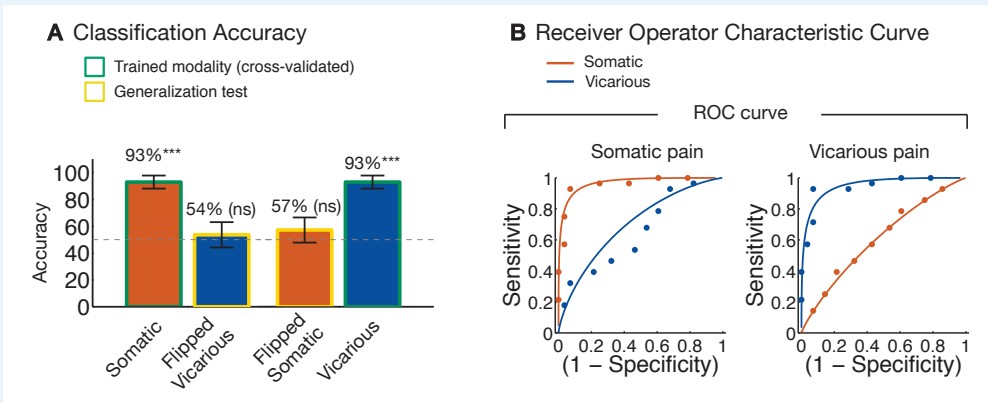

**Appendix 1—figure 6.** Upper limb versus lower limb analyses with left-right flipped testing data. (**A**) Accuracy statistics for upper limb versus lower limb weight maps for somatic and vicarious pain; (**B**) Receiver Operating Characteristic (ROC) curves for two-choice forced alternative tests for somatic and vicarious pain and high and low stimulations for each pain type.

## Extended discussion

A question that applies to both previous work and the current work is whether the neural similarities and differences across somatic and vicarious experience relate to pain or extraneous cognitive processing differences. What is at stake is whether the representations that *encode pain*, particularly in the dACC and aINS, are shared by vicarious pain. This question has not previously been fully examined, in part because previous studies have identified patterns that discriminate painful versus neutral stimuli but not those that predict pain experience with meaningfully large effect sizes [e.g., (*Corradi-Dell'Acqua et al., 2011*; *2016*)]. Four unique features of our study help to address whether the dissociations are trivial versus meaningful in terms of pain representations. First, strong predictive accuracy for both types of experience suggests that the patterns were related to pain, rather than other extraneous processes. Second, strong accuracy in out-of-sample (for new individuals) prediction also indicates that our study was highly powered to detect patterns that track experience, providing a meaningful test of shared representations. Third, lack of correlations between the NPS, VPS, and patterns predictive of rejection and negative emotion further suggest that these patterns are not sensitive to general emotional arousal. And fourth, somatotopy concerns brain organization *within* sensory modalities, which is expected to be similar if vicarious pain activates somatic pain representations. However, a full characterization of pain representations requires a long, multi-study program of research that is beyond the scope of this manuscript.

## Implications for pain representation

The critical issue addressed in this work is how somatic and vicarious pain representations are related, which raises important (and controversial) questions about what constitutes a neural representation of mental constructs such as pain (*Davis and Poldrack, 2013*; *Haxby et al., 2014*; *Panzeri et al., 2015*). One question concerns the level of analysis, as fMRI activity necessarily integrates across populations of many neurons, and whether single-neuron activity might provide a different and perhaps privileged picture. A second, related question concerns whether various forms of pain are represented by localized activity (in voxels or single neurons) or distributed patterns across populations. For example, *Hutchison et al., 1999* found that a subset of human dACC neurons (11/68, 16%) responds to noxious events, and a fraction of those neurons (1/11, 9%) also responded to observation of pain, albeit more weakly. The

overlapping neurons could indicate shared 'pain representation.' Alternatively, however, they may reflect engagement of common processes that are distinct from pain—for example, engagement of attention, which is known to activate dACC neurons. In addition, pain may be encoded not in single neurons, but in population vectors that relate to the pattern across many neurons, and even across macroscopic brain regions.

One way to establish a provisional representation is to characterize the sufficient basis for predicting related outcomes. Our study is grounded on the supposition that for a neural measure to serve as a representation of any particular type of pain, it should a) respond strongly to that type of pain (be *sensitive*); b) not respond to things that are similar but acknowledged to be distinct from pain (be *specific*); and c) generalize across multiple instances with different superficial properties (*Woo and Wager, 2015*). Only to the degree that the criteria are met does it become meaningful to test the relationship among the constructs. Previous studies have compared patterns for high versus low pain, but their results do not fulfill any of these three criteria. By these criteria, no single method (e.g., single-neuron approaches) are privileged as a measure, though some techniques may turn out to have much better measurement properties than others.

Our study takes several steps towards fulfilling these criteria, but also leaves much more work to be done. First, we identify patterns that are highly sensitive (and powered) to predict the magnitude of somatic and vicarious pain experience in out-of-sample individuals, and replicate those effects. Second, the NPS's specificity has been tested more broadly across datasets (*Chang et al., 2015*; *Woo et al., 2014*; *2015*), whereas we demonstrate specificity of the VPS here only with respect to somatic pain and anticipation (for somatic and vicarious pain), with no responses to any of these conditions. However, testing specificity is a multi-study process that ultimately requires examining many psychological conditions. The NPS and VPS have not yet been tested on all of the possible cognitive processes that may engage one of them or the other for reasons unrelated to pain (e.g., engagement of attention), though establishing patterns that predict pain experience helps to minimize influences of extraneous processes. Third, in Study 2 we demonstrate generalizability of the NPS across several types of somatic pain (pressure and mechanical), and we are actively examining the generalizability of the VPS to other empathy-related paradigms (e.g., 'cued empathy'). Assessing the full scope of what processes these patterns are sensitive to is an open-ended process, and a work in progress. However, the results thus far demonstrate promise that both patterns are useful as provisional representations of somatic and vicarious pain.

## Identifying shared representations for somatic and vicarious pain

Our findings do not suggest that there are *no* shared processes common to somatic and vicarious pain. Previous studies have investigated the shared representation question, and variously concluded that there is support for both shared (*Bruneau et al., 2013*; *Corradi-Dell'Acqua et al., 2011*; *Corradi-Dell'Acqua et al., 2016*) and distinct (*Bruneau et al., 2013*; *Woo et al., 2014*) brain bases for pain-related and other affective events that were interpreted as activating somatic pain representations precisely because they activated the dACC and aINS.

However, it is now widely recognized that these regions are activated by a wide variety of tasks that do not involve pain at all (*Iannetti et al., 2013*; *Lindquist et al., 2012*; *Mouraux et al., 2011*; *Yarkoni et al., 2011*). Thus, neither dissociable activity patterns (*Bruneau et al., 2013*) nor shared ones (*Corradi-Dell'Acqua et al., 2011*; *2016*) can tell us whether pain representations are involved unless at least one of the patterns tested is strongly predictive of pain experience. The finding of shared representation in the aINS (*Corradi-Dell'Acqua et al., 2011*)—which we replicated here—does not imply that those shared representations are actually representations of pain. What is shared could reflect common demands on attention, salience, and other more general processes. Consistent with this view, *Corradi-Dell'Acqua et al., 2011* provided evidence that the common patterns they identified in the aINS were not specific to pain, but rather reflected some more general aspect of negative emotional experience.

## Separate modifiability for somatic and vicarious pain predictive patterns

Our study tested the separate modifiability of somatic and vicarious pain patterns (*Sternberg, 2001*; *Woo et al., 2014*), which can rule out common influences of shared demands on attention, 'salience,' 'arousal,' and other more general processes. In brief, the logic is that if any hypothetical common process underlies both patterns, then both patterns should be affected by manipulations of the common process in consistent ways (*Sternberg, 2001*). In our results, two independent manipulations—of somatic pain intensity and vicarious pain intensity—differentially affect two unrelated patterns. Manipulating somatic pain influenced somatic pain representations (the NPS) but not vicarious pain representations (the VPS), and manipulating vicarious pain intensity influenced the VPS but not the NPS, demonstrating separate modifiability, in spite of comparable intensity ratings for somatic and vicarious pain. Furthermore, separate modifiability was found in both between-participant and within-participant whole brain analyses, and in the aINS.

This pattern of findings is logically incompatible with common-process accounts: If somatic and vicarious pain manipulations increased arousal (which might be reflected in the dACC or aINS), for example, and the NPS and VPS responses reflected arousal, then the vicarious pain manipulation should have increased activity in the NPS and vice versa. Because there was virtually no cross-prediction of either pattern, this implies that the signature diagnostic of each pain experience is independent and thus does not share overlapping processes. The lack of common patterns (cross-prediction) or shared representations for pain as compared with the findings of Corradi-Dell'Acqua and colleagues (*Corradi-Dell'Acqua et al., 2011*), or findings from other non-pain related domains such as listening to or executing actions (*Etzel et al., 2008*) can be interpreted alongside the strong positive findings. We demonstrate that the shared representation is likely to be much smaller, perhaps an order of magnitude smaller, than the strength of the within-modality effects. We also account for non-specific factors (e.g., arousal and salience) by training on pain experience and demonstrating separate modifiability. Thus, our results strongly suggest that the patterns identified in this study reflect distinct affective processes rather than general processes common to somatic and vicarious pain, such as emotional arousal.

## Towards signatures for multiple affective experiences

This paper presents a whole-brain multivariate signature that tracks pain empathy and can be applied prospectively to new individuals. We also applied the pattern to an independent replication dataset, which showed that the VPS dissociates somatic versus vicarious pain, and included two additional, independent datasets that confirm the generalizability of the NPS across pain modalities (heat, shock, and pressure), and confirms that the NPS is distinct from the VPS. Thus, these findings can be used, tested, and extended by many groups, allowing us to build a cumulative neuroscience of affect across laboratories.

Importantly, our results not only show that the neural bases of somatic and vicarious pain experiences are distinct, they also provide preliminary evidence that they are distinct from social rejection (*Woo et al., 2014*), another type of affective experience thought to be encoded in 'affective pain' systems. Though vicarious pain and social rejection share common involvement of some brain regions, such as the dorso-medial prefrontal cortex, they largely involve distinct regions even at the macroscopic scale. The social rejection pattern identified in *Woo et al. (2014)* involves a distinct set of regions associated with emotion generation [e.g., (*Lindquist et al., 2012*)], including the ventro-lateral prefrontal cortex, medial thalamus, and supplementary motor cortex. Thus, vicarious pain, somatic pain, and social rejection are distinct classes of affective experience, with distinct neural signatures that we can now identify and distinguish empirically.

