## [Decision Letter]

Thank you for submitting your article "Somatic and vicarious pain are represented by dissociable multivariate brain patterns" for consideration by *eLife*. Your article has been reviewed by Peggy Mason, with Jody Culham as the Senior Editor.

Because your original submission indicated that this manuscript had already undergone substantive reviews at two other high-profile journals (including six external reviewers altogether and two rounds of review at one of the journals), our editorial team carefully examined those reviews and your responses to them. In addition, your manuscript was thoroughly reviewed and evaluated by a Reviewing Editor (Peggy Mason). In our view, this mode of evaluation was the optimal use of the time and energy of reviewers, authors and editors. Ultimately, we are pleased that this approach allowed us to expedite the review process for your work by taking into consideration the considerable efforts you had already invested in addressing past reviewers' and editors' concerns satisfactorily.

This is a very important paper. It tackles a finding that in no short order became dogma and even slipped into popular 'fact.' The authors clearly demonstrate that the conflation of the brain representations of self and other pain is not warranted and that has been 'verified' without the simple but telling tests that the authors use in this paper. In particular, the authors compare 3 different intensities and 2 different locations and then toss in two additional modalities at single intensity and location. The work is compelling and shows that the somatic pain signature is different from the vicarious pain one. The important take-home message is (in the authors words) that "vicarious pain does not involve re-activation of somatosensory representations. Rather, body site-specific representations of vicarious pain may be accomplished using the mPFC 'mentalizing' system and perhaps other ideomotor systems."

The reviewing editor raised the following point that should be addressed in a revision:

Could the authors discuss in a concise way why they think that previous studies have come up with a different answer? While I am convinced by the data, I am puzzled by the long record of studies concluding the opposite. This may be in the Discussion already but if it is, it is not concise or punchy and does not stand out. Consequently, I did not see such a treatment although I was looking for it.

---

## [Author Response]

Could the authors discuss in a concise way why they think that previous studies have come up with a different answer? While I am convinced by the data, I am puzzled by the long record of studies concluding the opposite. This may be in the Discussion already but if it is, it is not concise or punchy and does not stand out. Consequently, I did not see such a treatment although I was looking for it.

This is an important request, as we think it is critical to be as clear as possible why our conclusions differ from those of many previous studies. We have added an additional section to the Discussion to address this, which is reproduced here:

“Most previous studies of pain empathy point out its similarity with somatic pain, and it is thus widely believed that the two experiences rely on the same systems. Why are our findings and conclusions different? There are three main reasons.

[…]

Based on our findings, we infer that the overlapping activation in dACC, aINS, and other areas is not related to shared pain experience. Interestingly, on close reading, the few previous multivariate pattern-based studies agree broadly with this interpretation. The brain patterns they identified as shared across somatic and observed pain were not specific to ‘pain,’ as these patterns were also activated by other, non-painful types of negative affect (Corradi-Dell’Acqua, Tusche, Vuilleumier, & Singer, 2016; Zaki et al.et al., 2016). As in our study, in these studies what is shared does not seem to be particular to pain per se.”